# Adaptive and Multi-scale Affinity Alignment for Hierarchical Contrastive Learning

**Jiawei Huang**[1,2]    **Minming Li**[2]    **Hu Ding**[1]*

[1]School of Computer Science and Technology, University of Science and Technology of China
[2]Department of Computer Science, City University of Hong Kong
{hjw0330, huding}@mail.ustc.edu.cn, minming.li@cityu.edu.hk

## Abstract

Contrastive self-supervised learning has emerged as a powerful paradigm for extracting meaningful representations without labels. While effective at capturing broad categorical distinctions, current methods often struggle to preserve the fine-grained and hierarchical relationships inherent in real-world data. From the perspective of semantic alignment, conventional contrastive learning aligns representations to semantic structure at a global level, treating the entire embedding space uniformly and frequently overlooking rich local structural information. In this paper, we propose *Adaptive Multi-scale Affinity alignment (AMA-alignment)*, a framework that introduces localized contrastive objectives and a dynamic multi-scale optimization strategy to adaptively identify and refine poorly aligned regions within the embedding space. Although our model is inherently more complex due to its *multi-scale* and *adaptive* design, we provide the theoretical guarantees indicating that its convergence rate remains comparable to that of standard smooth non-convex optimization. We conduct a set of experiments on diverse benchmarks to show that AMA-alignment can effectively preserve hierarchical structure; moreover, AMA-alignment also outperforms existing contrastive methods on a range of downstream tasks.

## 1 Introduction

Unsupervised/self-supervised learning has revolutionized the field of representation learning, achieving remarkable results across various domains with limited labeled data [1]. These approaches uncover intrinsic data relationships, facilitating the learning of meaningful representations without supervision. Among these methods, *contrastive learning (CL)* has proved to be a particularly effective framework. By leveraging data augmentations to generate positive and negative sample pairs, CL trains models to distinguish between similar instances (positive pairs) and dissimilar ones (negative pairs) [7, 51]. Beyond this discriminative objective, CL can also be interpreted as an "**alignment**" mechanism, encouraging the learned representations to reflect the semantic relationships implicitly defined by the data augmentation process [50, 48].

Despite its success in capturing high-level categorical distinctions, CL often struggles to preserve the fine-grained and hierarchical semantic relationships inherent in real-world data [1]. Such relationships are critical for downstream tasks where subtle variations carry significant semantic differences. For instance, while a CL model trained on the DeepFashion dataset [33] may correctly distinguish high-level categories like upper- and lower-body garments, it often struggles with finer-grained distinctions, failing to differentiate between t-shirts, jackets, sweaters, blouses, and vests.

To obtain fine-grained representations, recent advancements in CL mainly focus on designing stronger augmentation strategies. These methods construct more informative positive and negative sample

---

*Corresponding author.

39th Conference on Neural Information Processing Systems (NeurIPS 2025).

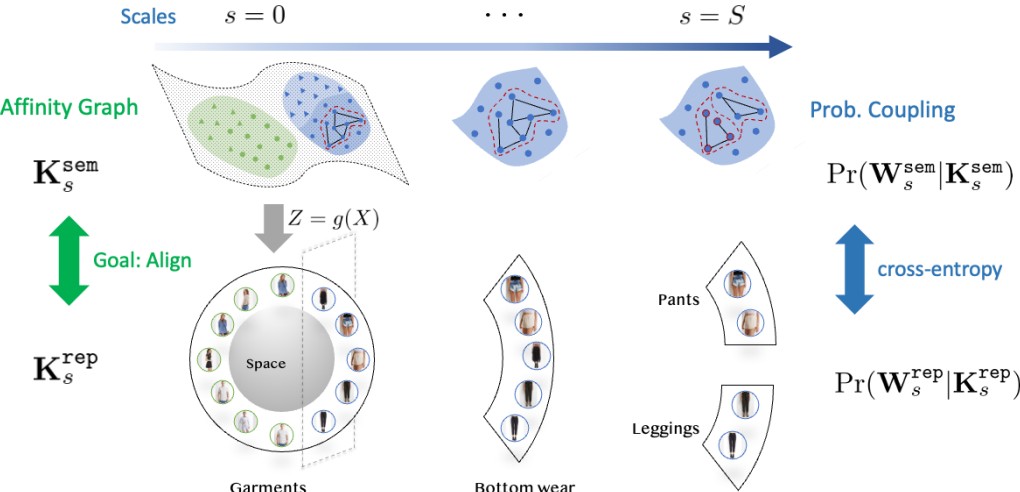

Figure 1: Illustration of multi-level contrastive learning. An ideal scenario where the learned representations exhibit a hierarchical structure that captures both broad categorical distinctions and fine-grained semantic relationships. Our approach identifies and optimizes local worst-case regions (highlighted in red) where representation affinity deviates significantly from the semantic affinity. Here, the affinity matrices $\mathbf{K}_s^{\text{sem}}$ and $\mathbf{K}_s^{\text{rep}}$ represent pairwise semantic and representation relationships, respectively, as defined in (3) and (4).

pairs by incorporating task-specific cues or leveraging multi-view inputs [32, 54, 12]. For example, Feng et al. [12] leveraged diverse modalities and perspectives of chemical molecular data to construct more effective positive and negative pairs. Additionally, Robinson et al. [43] proposed a method for sampling hard negative examples, improving the quality of negative sample selection.

Our intuition for addressing these challenges originates from hierarchical clustering, as it also aims to capture structural relationships across multiple levels of granularity. Recent theoretical advances [55, 50, 19, 9] also reveal fundamental connections between CL and traditional clustering methods. These studies show that contrastive objectives encourage representations to distribute uniformly on the unit sphere $\mathbb{S}^{d_g-1}$ and naturally form category-based clusters. In the ideal case, as illustrated in the lower half of Figure 1, such representations would reflect not only coarse categorical structure but also fine-grained semantic hierarchies, thus resembling hierarchical clustering. For example, sub-categories of lower-body clothing would be effectively clustered within the broader lower-body region. However, while existing CL methods perform well in capturing broad distinctions, they typically learn these semantic relationships at a global level, often failing to preserve the fine-grained relationships critical for downstream tasks.

To overcome this limitation, we seek to incorporate hierarchical structure into the CL process more directly. Hierarchical clustering [38, 39] naturally creates *multi-scale* region partitions using either top-down (divisive) or bottom-up (agglomerative) strategies. However, several fundamental obstacles hinder the direct application of conventional hierarchical clustering techniques in the CL context: **1)** Classical hierarchical clustering methods rely on distances that are directly accessible, whereas CL operates on an affinity graph $\mathbf{K}^{\text{sem}}$ (formally introduced in Sec. 2) whose edge weights are determined by a stochastic positive pair sampling process, making them nontrivial to compute directly. **2)** Unlike *non-parametric* clustering algorithms, which do not rely on learned feature mappings [19], CL trains a parameterized encoder $g_\theta(\cdot)$ to produce feature embeddings to *align* semantic structure $\mathbf{K}^{\text{rep}}$ (see (4)), therefore requires to redesign a more appropriate optimization strategy for $g_\theta(\cdot)$. **3)** In fine-grained alignment scenarios, different local regions of the embedding space often exhibit varying degrees of semantic misalignment [13]. Uniformly optimizing over the global affinity structure is inefficient; in other words, it should be more effective to prioritize regions that are poorly aligned with the underlying semantics. These limitations call for a region-aware approach that retains the hierarchical perspective while adapting to CL's characteristics.

**Our proposed model and contributions.** The above challenges motivate the development of our hierarchical region-aware optimization framework, which dynamically prioritizes poorly aligned regions across multiple semantic scales. We use $\{1, 2, ..., S\}$ to denote different scales, ranging from coarse to fine-grained. At each scale $s$, we partition the embedding space into a set of regions $\{R_r^s\}_{r=1}^{m^s}$, where $m^s$ is the number of regions at that scale. For each region $R_r^s$, we define a corresponding *local contrastive loss* $\mathcal{L}^s(\theta, R_r^s)$ (formally derived in Sec. 3.1) that measures how well the representation preserves semantic relationships within that specific region. Specifically, we formulate a new multi-scale optimization objective as follows:

$$\min_{\theta \in \Theta} \sum_{s=1}^{S} \max_{r \in [m^s]} \{\mathcal{L}^s(\theta, R_r^s)\}. \tag{1}$$

The inner maximization adaptively identifies regions with high loss values, while the outer minimization over $\theta$ focuses on reducing the sum of the regional losses. We name our method *adaptive multi-scale affinity alignment (AMA-alignment)* and highlight its key contributions:

- We introduce the new concept of *local affinity graph* and derive its associated *local contrastive loss*, specifically designed to align regional structures. We further propose a new perspective that integrates *distributionally robust optimization* with local contrastive learning to adaptively refine poorly aligned regions. This framework facilitates multi-scale representation learning and captures both coarse and fine-grained semantic structures.

- We provide a theoretical analysis of the proposed optimization objective and algorithm. In particular, even our optimization model is more complicated due to the endowed "multi-scale" and "adaptive" properties, the convergence rate is comparable to the standard smooth non-convex optimization.

- We evaluate AMA-alignment on a range of benchmark datasets, demonstrating its effectiveness in preserving semantic structure and excelling on challenging downstream benchmarks.

## 1.1 Other Related Works

Several studies have leveraged multi-level labels [25, 66] to learn hierarchical representations. These methods are supervised approaches that rely on annotated data across multiple levels. In contrast, our work focuses on unsupervised learning, consistent with conventional settings of CL algorithms. Mo et al. [36] introduced a method that employs two types of contrastive losses at both shallow and deep network layers, enhancing the quality of learned representations. Additionally, other works [14, 59] have explored embedding representations in hyperbolic space to better capture hierarchical relationships. More recently, several studies [32, 54, 12] have focused on refining data augmentation strategies to more effectively encode semantic information, either explicitly or implicitly. Our study demonstrates that by more effectively utilizing the semantic information inherently encoded in augmentations, the model can capture fine-grained semantic structures more accurately through improved optimization techniques. A more detailed discussion of related works is deferred to Appendix A.

## 2 Preliminaries

### 2.1 Formulation for Contrastive Learning (CL)

Let $\bar{X}$ denote the input dataset sampled from the natural data distribution $\mathcal{P}_{\bar{X}}$, and let $A(\cdot)$ be a randomized data augmentation operator (*e.g.,* cropping, color jittering). For each $\bar{x} \in \bar{X}$, we define the conditional distribution $\Pr_A(\cdot|\bar{x})$ as the probability density over the data space $\mathcal{X}$ induced by applying $A(\cdot)$ to $\bar{x}$. That is, $\Pr_A(x|\bar{x})$ denotes the likelihood of obtaining $x$ as an augmented view of $\bar{x}$. By applying $A(\cdot)$ to all samples in $\bar{X}$, we obtain the augmented data distribution $\mathcal{P}_X$, whose support set is denoted as $X$. Without loss of generality, we assume $X$ is a finite set of size $N$; if $X$ is infinite, the summations in our analysis can naturally generalize to integrals.

We model the network as $\Gamma = h \circ g$, where $g(\cdot) : \mathcal{X} \to \mathcal{Z}$ is the encoder mapping input data to a representation space $\mathcal{Z} \subset \mathbb{R}^{d_g}$, and $h : \mathcal{Z} \to \mathcal{Y}$ is a task-specific classifier. The encoder $g(\cdot)$ is parameterized by $\theta \in \Theta$, where $\Theta$ denotes the parameter space. Contrastive Learning (CL) aims to learn discriminative representations by pulling semantically similar instances closer and pushing

dissimilar ones apart in the embedding space [51, 58, 63]. Following standard CL approaches, we adopt the *Information Noise-Contrastive Estimation (InfoNCE)* objective [7, 18]:

$$\mathcal{L} = \mathop{\mathbb{E}}_{\bar{x} \sim \mathcal{P}_{\bar{X}}} \left[ \ell_{\mathrm{cl}}\left(\bar{x}\right) \right], \quad \text{where } \ell_{\mathrm{cl}}(\bar{x}) = - \mathop{\mathbb{E}}_{x_i, x_j \sim \mathrm{Pr}_{\mathrm{A}}(\cdot | \bar{x})} \left[ \log \frac{\exp\left(g(x_i)^\top g(x_j)/\tau\right)}{\sum_{k \neq i} \exp\left(g(x_i)^\top g(x_k)/\tau\right)} \right]. \quad (2)$$

Here $\tau > 0$ is the temperature parameter and $x_k$ is a negative sample from $X$. Let $Z = \{z_i\}_{i=1}^N \subset \mathcal{Z}$ denote the set of feature embeddings for the augmented dataset $X$, where each $z_i = g(x_i)$ corresponds to a data point $x_i \in X$.

## 2.2 Alignment via Probabilistic Graph Coupling

We reinterpret the contrastive objective in (2) as aligning two graphs—one defined by semantic affinities and the other by representation similarities. This insight motivates our key idea: generalizing the global graph alignment to a hierarchical framework that captures structural correspondences across multiple spatial scales.

**Definition 1** (Semantic Affinity Matrix)**.** The semantic affinity matrix $\mathbf{K}^{\mathtt{sem}} \in \mathbb{R}_+^{N \times N}$ over $X$ quantifies the pairwise semantic proximity through augmentation co-occurrence. Specifically, for any $x_i, x_j \in X$, its $(i, j)$-th entry is defined as:

$$\mathbf{K}_{ij}^{\mathtt{sem}} \triangleq \mathbb{E}_{\bar{x} \sim \mathcal{P}_{\bar{X}}} \left[ \mathrm{Pr}_{\mathrm{A}}(x_i | \bar{x}) \cdot \mathrm{Pr}_{\mathrm{A}}(x_j | \bar{x}) \right]. \quad (3)$$

This matrix quantifies the joint probability that $x_i$ and $x_j$ are both generated from the same underlying sample $\bar{x}$, thereby encoding their semantic relatedness under data augmentation.

**Definition 2** (Representation Affinity Matrix)**.** The representation affinity matrix $\mathbf{K}^{\mathtt{rep}} \in \mathbb{R}_+^{N \times N}$ over $Z$ measures pairwise similarities in the embedding space. For any $x_i, x_j \in X$:

$$\mathbf{K}_{ij}^{\mathtt{rep}} \triangleq \mathtt{Ker}(z_i, z_j), \quad \text{where } z_i = g(x_i), \; z_j = g(x_j), \quad (4)$$

where $\mathtt{Ker} : \mathcal{Z} \times \mathcal{Z} \to \mathbb{R}_+$ is a translation-invariant kernel function (*e.g.,* Gaussian kernel) [55].

$\mathbf{K}^{\mathtt{sem}}$ and $\mathbf{K}^{\mathtt{rep}}$ define two weighted graphs: the former reflects semantic relationships via augmentation, while the latter encodes geometric proximities in representation space. Prior research [55, 50] has shown that CL aligns these graphs by learning representations that preserve semantic relationships.

**Probabilistic Graph Coupling.** To formalize this alignment process, we adopt the Probabilistic Graph Coupling framework [55], which interprets CL as minimizing divergence between distributions over subgraphs sampled from $\mathbf{K}^{\mathtt{sem}}$ and $\mathbf{K}^{\mathtt{rep}}$. These distributions reflect the structural properties of the underlying graphs, and CL aligns them by aligning their subgraph statistics.

We restrict attention to binary directed graphs $\mathbf{W}$ with exactly one outgoing edge per node—mirroring the single-positive sampling mechanism used in contrastive learning:

$$\mathcal{S}^{\mathbf{W}} \triangleq \{\mathbf{W} \in \{0, 1\}^{N \times N} \,|\, \forall i \in [N], \textstyle\sum_{j=1}^N \mathbf{W}_{ij} = 1\}, \quad (5)$$

where $\mathbf{W}_{ij} = 1$ denotes a directed edge from node $x_i$ to $x_j$. We impose this constraint as it mimics the positive pairing mechanism in contrastive learning, where each data point selects exactly one positive sample. Thus the InfoNCE objective can be reinterpreted as minimizing the divergence between subgraph distributions over the constraint (5). We summarize the core idea below, and provide a detailed discussion in Appendix B.

Given an affinity matrix $\mathbf{K} \in \{\mathbf{K}^{\mathtt{sem}}, \mathbf{K}^{\mathtt{rep}}\}$, as defined in Def. 1 or 2), the probability of sampling a subgraph $\mathbf{W}$ under the topological constraints specified in (5) is given by:

$$\mathrm{Pr}(\mathbf{W}|\mathbf{K}) \propto \Omega(\mathbf{W}) \textstyle\prod_{(i,j) \in [N]^2} (\mathbf{K}_{ij})^{\mathbf{W}_{ij}}, \quad (6)$$

where $\Omega(\mathbf{W}) \triangleq \prod_{i=1}^N \mathbb{1}\left(\sum_j \mathbf{W}_{ij} = 1\right)$ enforces the topological constraint, ensuring that each node has exactly one outgoing edge. The core objective of contrastive learning can thus be reformulated as minimizing the cross-entropy between the subgraph distributions induced by $\mathbf{K}^{\mathtt{sem}}$ and $\mathbf{K}^{\mathtt{rep}}$:

$$\mathbb{E}_{\mathbf{W}^{\mathtt{sem}} \sim \mathrm{Pr}(\cdot | \mathbf{K}^{\mathtt{sem}})} - \left[ \log \mathrm{Pr}(\mathbf{W}^{\mathtt{rep}} = \mathbf{W}^{\mathtt{sem}} \mid \mathbf{K}^{\mathtt{rep}}) \right]. \quad (7)$$

As shown in [50], minimizing (7) is equivalent to the InfoNCE loss in (2). This equivalence formally links CL with probabilistic graph alignment.

# 3 Method

This section introduces our hierarchical contrastive learning framework. We begin with a high-level overview and then detail its two key components.

**Overview.** Our method performs contrastive alignment across multiple spatial granularities in a unified and synchronous manner, as formalized in equation (1) (Sec. 1). It operates simultaneously on: *(i) global structure*, by preserving high-level semantic consistency through large-scale alignment; and *(ii) local structure*, by enhancing fine-grained discriminability within local neighborhoods. This multi-scale co-optimization leads to robust representation learning that spans from global categories to subtle intra-class variations. The framework comprises two main technical components:

1. **Local Alignment (Sec. 3.1):** We define local regions in the embedding space $\mathcal{Z}$ and align corresponding semantic and representation affinity graphs using a novel local contrastive loss $\mathcal{L}^s$ (12), which is derived by the Probabilistic Graph Coupling technique.

2. **Adaptive Refinement (Sec. 3.2):** We introduce the adaptive multi-scale alignment algorithm that aggregates local losses across scales using a distributionally robust optimization (DRO) scheme, which dynamically prioritizes challenging regions.

## 3.1 Local Contrastive Learning Loss

We first describe how to locally align semantic and representation affinities. Extending the affinity-based interpretation of the InfoNCE loss (2), we develop a localized version that aligns affinity graphs within small, spatially defined regions.

**Local affinity graph.** In contrastive learning, the input space $\mathcal{X}$ is mapped and normalized into a unit hypersphere embedding space $\mathcal{Z} \subset \mathbb{S}^{d_g-1} \subset \mathbb{R}^{d_g}$ via a trainable encoder $g(\cdot)$ [58]. To capture structure at multiple spatial resolutions, we introduce $S$ levels of granularity. Each scale $s$ is associated with an angular radius $\iota^s > 0$ that defines the size of local regions. At scale $s$, we define $m^s$ local regions $\mathcal{R}^s(\theta) = \{R^s(\theta, z_r^s)\}_{r=1}^{m^s}$, each centered at an anchor point $z_r^s \in \mathcal{Z}$. For brevity, we denote the $r$-th region at scale $s$ by $R_r^s$. Given anchor $z_r^s$ and radius $\iota^s$, the local region is defined as:

$$R_r^s = \left\{ z \in \mathcal{Z} \mid \angle(z, z_r^s) \leq \iota^s \right\}, \tag{8}$$

where $\angle(\cdot, \cdot)$ is the angular distance (*i.e.,* the arc-cosine of the normalized dot product). This region captures points most similar to the anchor under the current embedding, naturally adapting to local data geometry. To build intuition, a coarse scale (large $\iota^s$) may group together all clothing items, whereas a finer scale (small $\iota^s$) could isolate specific categories such as jeans, skirts, or shorts.

Let $X_{R_r^s} \subseteq X$ denote the subset of samples whose embeddings fall inside region $R_r^s$, indexed by $I_{R_r^s}$. That is, $X_{R_r^s} = X[I_{R_r^s}]$ with $|X_{R_r^s}| = N'$. We define the *local affinity matrices* $\mathbf{K}_s^{\text{sem}}, \mathbf{K}_s^{\text{rep}} \in \mathbb{R}_+^{N' \times N'}$ as submatrices of the global affinity matrices $\mathbf{K}^{\text{sem}}$ and $\mathbf{K}^{\text{rep}}$ (see Def. 1 and 2):

$$\mathbf{K}_s^{\text{sem}} \triangleq \mathbf{K}^{\text{sem}}[I_{R_r^s}, I_{R_r^s}], \quad \mathbf{K}_s^{\text{rep}} \triangleq \mathbf{K}^{\text{rep}}[I_{R_r^s}, I_{R_r^s}]. \tag{9}$$

Since $\mathbf{K}^{\text{sem}}$ and $\mathbf{K}^{\text{rep}}$ are symmetric, the index set $I_{R_r^s}$ applies to both rows and columns. When the specific region $r$ is not of primary interest, we omit the region index for notational simplicity.

**Local contrastive loss.** To align $\mathbf{K}_s^{\text{sem}}$ and $\mathbf{K}_s^{\text{rep}}$, we again employ the probabilistic graph coupling framework. Similar to (7), this alignment is achieved by minimizing the cross-entropy between the distributions of subgraphs $\mathbf{W}_s^{\text{sem}}$ and $\mathbf{W}_s^{\text{rep}}$ sampled from $\mathbf{K}_s^{\text{sem}}$ and $\mathbf{K}_s^{\text{rep}}$, respectively. These subgraphs follow the constraint defined in (5), ensuring each node has exactly one outgoing edge. This leads to the following distribution over sampled subgraphs:

**Proposition 3.1.** *Let* $\mathbf{W}_s$ *be a subgraph sampled from* $\mathbf{K}_s$ *(represents either* $\mathbf{K}_s^{\text{sem}}$ *or* $\mathbf{K}_s^{\text{rep}}$*) under the constraint in* (5). *Then:*

$$\Pr(\mathbf{W}_s | \mathbf{K}_s) \propto \Omega(\mathbf{W}_s) \prod_{(i,j) \in [N']^2} (\mathbf{K}_{s,ij})^{\mathbf{W}_{s,ij}}, \tag{10}$$

*where* $\Omega(\mathbf{W}_s) \triangleq \prod_i \mathbb{1}(\sum_j \mathbf{W}_{s,ij} = 1)$ *enforces the one-outgoing-edge constraint* (5).

Equation (10) serves as the localized counterpart of the global formulation in (6), extending the probabilistic graph coupling to the region-specific setting. To further formalize the concept of learning within localized regions, we introduce a *restricted augmentation distribution* that limits sampling to examples whose representations lie within a specific region:

$$\Pr_r^s(x|\bar{x}) = \begin{cases} \frac{\Pr_A(x|\bar{x})}{\chi_r^s}, & \text{if } g(x) \in R_r^s, \\ 0, & \text{otherwise,} \end{cases} \tag{11}$$

where the normalization constant $\chi_r^s = \int_{x \in \mathcal{X}} \mathbb{1}_{g(x) \in R_r^s} \Pr_A(x|\bar{x}) \mathrm{d}x$ ensures that $\Pr_A(\cdot|\bar{x})$ defines a valid probability distribution. This formulation constrains the augmentation process to operate only within the local regions $R_r^s$.

Similar to the InfoNCE loss, following the probabilistic graph coupling framework, we minimize the cross-entropy between the subgraph distributions over the local affinity matrices $\mathbf{K}_s^{\mathtt{sem}}$ and $\mathbf{K}_s^{\mathtt{rep}}$, and derive the *local contrastive loss* as shown in (12):

$$\mathcal{L}^s = \mathbb{E}_{\bar{x} \sim \mathcal{P}_{\bar{X}}} \left[ \ell_{\mathrm{cl}}^s(\bar{x}) \right], \quad \text{where} \quad \ell_{\mathrm{cl}}^s(\bar{x}) = - \mathbb{E}_{x_i, x_j \sim \Pr_r^s(\cdot|\bar{x})} \left[ \log \frac{\exp(z_i^\top z_j / \tau)}{\sum_{k \neq i} \mathbb{1}_r^s(x_k) \exp(z_i^\top z_k / \tau)} \right]. \tag{12}$$

Here, $\tau$ is the temperature parameter, $z_i = g(x_i)$ and $z_j = g(x_j)$ are embeddings produced by the encoder $g$, and $\mathbb{1}_r^s(x_k) = \mathbb{1}_{g(x_k) \in R_r^s}$ ensures negatives are drawn from the local region. We have the following lemma:

**Lemma 3.2.** *Minimizing the cross-entropy between the subgraph distributions induced by $\mathbf{K}_s^{\mathtt{sem}}$ and $\mathbf{K}_s^{\mathtt{rep}}$, under the structural constraint in* (5)*, is equivalent to minimizing the* (12)*.*

This lemma confirms that the local loss in (12) is theoretically grounded in the same probabilistic framework as the global contrastive loss (2), but adapted to align affinity structure within localized regions. A full proof is provided in Appendix C.

**Remark 1.** The formulation of $\mathcal{L}^s$ is non-trivial. A naive approach might define a local contrastive loss by restricting positive pairs to lie within the local region $R_r^s$, such as:

$$\tilde{\ell}_{\mathrm{cl}}^s(\bar{x}) = - \mathbb{E}_{x_i, x_j \sim \Pr_r^s(\cdot|\bar{x})} \left[ \log \frac{\exp(z_i^\top z_j / \tau)}{\sum_{k \neq i} \exp(z_i^\top z_k / \tau)} \right]. \tag{13}$$

However, the sampling distribution implied by (13), $\frac{\exp(z_i^\top z_j / \tau)}{\sum_{k \neq i} \exp(z_i^\top z_k / \tau)}$, does not reflect the true edge sampling probability in the underlying subgraph, *i.e.,* $\Pr(\mathbf{W}_{s,ij}^{\mathtt{rep}} = 1) \neq \frac{\exp(z_i^\top z_j / \tau)}{\sum_{k \neq i} \exp(z_i^\top z_k / \tau)}$. Therefore, using (13) does not ensure the alignment between local and global affinities, violating the alignment principle of CL. For example, this formulation would distort the global structure by inappropriately pushing all representations $z' \notin R_r^s$ maximally away from representations $z \in R_r^s$.

## 3.2 Adaptive Multi-scale Optimization for Affinity Alignment

Building on the local contrastive loss (12) in Sec. 3.1, we now describe how these losses are integrated into a unified optimization process that emphasizes difficult regions across multiple semantic scales. Our key insight is that not all regions require equal attention during training. In real-world datasets, semantic misalignment often occurs heterogeneously—certain neighborhoods in the embedding space may already adequately preserve semantic relationships, while others exhibit significant distortions. For instance, in fashion datasets, the boundary between "formal shirts" and "casual shirts" might be more challenging to delineate than the distinction between "shirts" and "pants", and thus it requires more effort to design an appropriate optimization strategy.

We formalize this intuition through a novel adaptive multi-scale optimization framework that dynamically prioritizes poorly aligned regions while maintaining coordination across scales. Our objective, introduced in equation (1), employs a nested min-max structure that adaptively allocates the focus of current computational stage towards the most challenging regions. For analytical clarity and to facilitate tractable optimization, we present the following equivalent reformulation of (1) using a convex combination of region-wise losses:

$$\min_{\theta \in \Theta} \sum_{s=1}^{S} \max_{\mathbf{q}^s \in \Delta_{m^s}} \sum_{r=1}^{m^s} \mathbf{q}_r^s \cdot \mathcal{L}^s(\theta, R_r^s). \tag{14}$$

where $\mathbf{q}^s = (q_1^s, ..., q_{m^s}^s)$ lies within the probability simplex $\Delta_{m^s} \triangleq \{\mathbf{q}^s \mid \sum_{i=1}^{m^s} q_i^s = 1, q_i^s \geq 0\}$ and represents the adaptive weighting over regions at scale $s$. The inner maximization identifies regions with high loss values, while the outer minimization over $\theta$ directs learning to reduce regional losses. This formulation is partially inspired by the distributionally robust optimization (DRO) [45, 65], and we leave the discussion on DRO to Sec. 3.3.

As introduced in Sec. 3.1, our framework operates across $S$ levels of spatial granularity. Each scale $s$ is characterized by its angular radius $\iota^s$, which determines the size of local regions. At each scale, we define $m^s$ regions $\mathcal{R}^s(\theta) = \{R_r^s\}_{r=1}^{m^s}$ partitioning the embedding space $\mathcal{Z}$ according to (8). Finer scales (small $\iota^s$) yield more focused regions, while coarser scales (large $\iota^s$) group semantically broader patterns. This hierarchical partitioning can be efficiently implemented through spherical $k$-means clustering on the unit hypersphere. Due to the limited space, we provide our implementation details in Appendix F.

However, directly optimizing this min-max objective (14) brings two key challenges: (1) due to the non-convexity of $\mathcal{L}^s(\theta, R_r^s)$ w.r.t. $\theta$ and the nested min-max structure, the standard convergence guarantees cannot be easily obtained; (2) the inner maximization over $\mathbf{q}^s$ may lead to degenerate solutions that excessively concentrate on a single region, resulting in unstable training dynamics. To address these challenges, we introduce an entropy regularization term and reformulate the objective in (14) as:

$$F(\theta) = \min_{\theta \in \Theta} \sum_{s=1}^{S} \Big( \max_{\mathbf{q}^s \in \Delta_{m^s}} \sum_{r=1}^{m^s} \mathbf{q}_r^s \cdot \mathcal{L}^s(\theta, R_r^s) - \rho \mathbf{KL}(\mathbf{q}^s) \Big), \tag{15}$$

where $\mathbf{KL}(\mathbf{q}^s) = \sum_{r=1}^{m^s} \mathbf{q}_r^s \log\big(m^s \mathbf{q}_r^s\big)$ is the KL divergence between $\mathbf{q}^s$ and the uniform distribution, and $\rho > 0$ is a regularization parameter controlling the strength of the entropic penalty.

**Benefits of KL Term in** (15). The KL regularization term $\rho \sum_{i=1}^{m^s} \mathbf{q}_r^s \log(m^s \mathbf{q}_r^s)$ in (15) serves three critical functions: **1)** It ensures strong concavity in $\mathbf{q}^s$, which is essential for proving convergence in the non-convex setting for our AMA-alignment algorithm, as established in Theorem D.10. **2)** It prevents the weight distribution from collapsing to the simplex boundary, stabilizing the optimization dynamics and improving convergence behavior. **3)** It enables a closed-form update for $\mathbf{q}^s$ via exponential weights (see (17)), avoiding iterative gradient ascent and facilitating efficient implementation.

---

**Algorithm 1** Adaptive Multi-scale Affinity Alignment (AMA-alignment) for Hierarchical CL

---

**Input:** Dataset $X$, encoder $g(\cdot)$, scale radius $\{\iota^s\}_{s=1}^S$, iterations $T$, learning rate $\eta_{\theta,t}$, regularization $\rho$
**Output:** Optimized parameters $\theta$
 1: Initialize $\theta_0$ from base model
 2: Pre-train encoder $g(\cdot)$ with standard contrastive loss $\mathcal{L}$ for $N_0$ epochs
 3: Generate partitions $\{R_r^s\}_{r=1}^{m^s}$ for each scale $s$ (detailed in Appendix F)
 4: For each scale $s \in [S]$, initialize $\mathbf{q}^s = [1/m^s, \ldots, 1/m^s]^T$
 5: **for** $t = 1$ to $T$ **do**
 6:    **for** each scale $s \in [S]$ **do**
 7:       Compute region losses $\{\mathcal{L}^s(\theta_t, R_r^s)\}_{r=1}^{m^s}$
 8:       Compute optimal weights $\mathbf{q}^s(\theta_t)$ by (17)
 9:    **end for**
10:    Compute total gradient $\mathbf{g}_t$ by (16)
11:    Update encoder: $\theta_{t+1} = \theta_t - \eta_{\theta,t} \mathbf{g}_t$
12: **end for**
13: **return** $\theta_T$

---

**Sketch of Algorithm 1.** Our AMA-alignment method is outlined in Algorithm 1. The training begins with a pre-training phase using the global contrastive loss $\mathcal{L}$ for $N_0$ epochs, which can be interpreted as optimizing at the global scale ($s = 0$). After this initialization, the model proceeds to jointly optimize affinity alignment across multiple spatial scales, refining both global and local semantic structures. Each iteration involves two key steps: updating encoder parameters $\theta$ (Algorithm 1, lines $7-8$) and adaptively adjusting region weights $\mathbf{q}_t^s$ (Algorithm 1, line 10):

*Updating Encoder Parameters (θ):* The encoder parameters are updated via gradient descent at each iteration $t$. The gradient estimator $\mathbf{g}_t$ is computed using the current region weights $\mathbf{q}_t^s$:

$$\theta_{t+1} = \theta_t - \eta_{\theta,t}\mathbf{g}_t, \qquad \text{where } \mathbf{g}_t = \sum_{s=1}^{S}\sum_{r=1}^{m^s} \mathbf{q}_{t,r}^s \nabla\mathcal{L}^s(\theta_t, R_r^s). \tag{16}$$

Here, $\eta_{\theta,t}$ is the learning rate for the encoder parameters.

*Updating Region Weights ($\mathbf{q}^s$):* The region weights are then adaptively updated based on the new encoder parameter $\theta_{t+1}$. This step follows the standard *stochastic mirror ascent* procedure, which solves a regularized inner optimization problem as formulated in (36) (see Appendix D.1) and admits a unique closed-form solution given in (17). For notational simplicity, we define a time-dependent coefficient $\gamma_t = 1 + \rho, \eta_{\mathbf{q},t}$. The updated weight for each region $r$ is then given by

$$\mathbf{q}_{t+1,r}^s = \frac{(\mathbf{q}_{t,r}^s)^{1/\gamma_t}\exp\left(\frac{\eta_{\mathbf{q},t}}{\gamma_t}\mathcal{L}^s(\theta_{t+1}, R_r^s)\right)}{\sum_{j=1}^{m^s}(\mathbf{q}_{t,j}^s)^{1/\gamma_t}\exp\left(\frac{\eta_{\mathbf{q},t}}{\gamma_t}\mathcal{L}^s(\theta_{t+1}, R_j^s)\right)}. \tag{17}$$

where $\eta_{\mathbf{q},t}$ controls the adaptation rate of the region weights. This update mechanism automatically prioritizes challenging regions by assigning them higher weights. Regions with higher loss values receive greater attention in subsequent optimization steps, dynamically focusing computational resources on the most difficult regions.

**Theorem 3.3** (Convergence Rate (Informal)). *Under the smoothness assumption of the objective, suppose the learning rates are set as $\eta_{\theta,t} = \frac{\alpha}{\sqrt{t}}$ for parameter updates and $\eta_{\mathbf{q},t} = \frac{\eta_0}{t}$ for region weight updates, where $\alpha, \eta_0 > 0$ are constants. Then Algorithm 1 achieves the following convergence rate: $\frac{1}{T}\sum_{t=1}^{T}\mathbb{E}[\|\nabla F(\theta_t)\|^2] \leq \frac{C}{\sqrt{T}}$, where $F(\theta)$ is the overall DRO objective in (15) and $C$ is a constant independent of $T$.*

We analyze Algorithm 1 in a stochastic optimization setting, where both the region-level loss (lines 7–8) and the gradient $\mathbf{g}_t$ (line 10) are computed via stochastic approximations. To ensure stable training under this noise, we update the region weights $\mathbf{q}^s(\theta_t)$ recursively using a moving-average rule (17), rather than solving the inner maximization exactly—despite it admitting a closed-form. Under mild assumptions (e.g., bounded gradients/losses and the model smoothness), our algorithm converges at a rate comparable to standard smooth non-convex methods, despite the multi-scale structure and adaptive updates. The full proof of Theorem D.10 is provided in Appendix D.

### 3.3 Some More Discussions

**Discussion of relations to Group DRO methods [45]** Our approach is partly inspired by group DRO method but with following differences: **1) Hierarchical, data-driven grouping:** Our method dynamically forms groups based on local neighborhoods in the evolving embedding space rather than fixed, predefined partitions; **2) Region-specific loss:** We employ specialized losses derived from the affinity-based interpretation of InfoNCE, aligning representations based on semantic and geometric structure rather than generic empirical risks. **3) Non-convex convergence guarantees:** Our analysis establishes convergence in the challenging non-convex optimization settings typical of deep contrastive learning, unlike prior work [44, 65] that focused primarily on convex settings.

**Analyzing the Optimization Dynamics of AMA-alignment.** Previous theoretical work [52] interprets contrastive learning (CL) as a two-player coordinate-wise optimization game between model parameters and sample-pair weights—an interpretation that encompasses classical strategies like *hard negative mining*[43]. AMA-alignment generalizes this view into a hierarchical three-player optimization framework. In addition to the standard optimization over model parameters and pairwise affinities, we introduce a third axis: region-level importance weights that adaptively prioritize difficult semantic regions across multiple spatial scales. This additional layer of adaptivity enables AMA-alignment to capture both local semantic granularity and global representational structure. We formally analyze this three-level coordinate-wise optimization dynamics in Section E.

# 4 Experiments

We evaluate the proposed **AMA-alignment** framework on multiple benchmarks to assess its ability to capture fine-grained and hierarchical semantic structures. We compare against state-of-the-art contrastive learning (CL) baselines in both unsupervised and supervised settings.

## 4.1 Experimental Setup

**Datasets.** We experiment on diverse datasets with inherent hierarchical or fine-grained semantics: (1) **DeepFashion** [33], a fine-grained clothing dataset; (2) **iNaturalist** [56], with taxonomic labels; (3) **CIFAR-100** [27], grouped into superclasses; and (4) **ModelNet40** [61], a 3D object classification dataset. (5) **BuImg** [37], a breast cancer diagnosis ultrasound images dataset.

**Baselines.** We compare with popular CL methods including SimCLR [7], MoCo [8], HardNeg [43], and $\alpha$-CL [52]. In supervised settings, we compare with Cross-Entropy, SupCon [26], Guided-Proto [28], and HiMulConE [66]. Unless otherwise specified, we train ResNet-50 [17] as backbone. All models are implemented with PyTorch on a single NVIDIA RTX 6000 Ada GPU. The model is optimized using the AdamW optimizer [34]. Additional experimental details are in Appendix G.

## 4.2 Experimental Results

Table 1 shows the classification results. AMA-alignment consistently outperforms existing CL methods, with substantial gains on fine-grained benchmarks. On DeepFashion, we observe over **7%** improvement, highlighting AMA's ability to capture subtle semantics. Performance on iNaturalist also improves notably, especially in tail classes.

Table 1: Accuracy on downstream classification tasks. AMA-alignment significantly improves fine-grained performance.

| Method | DeepFashion | ModelNet | iNat | ImageNet |
|---|---|---|---|---|
| SimCLR [7] | 70.3 | 79.3 | 54.0 | 69.5 |
| MoCo-v2 [8] | 70.8 | 79.6 | 55.3 | 68.1 |
| HardNeg [43] | 70.9 | 79.8 | 55.8 | 70.4 |
| $\alpha$-CL [52] | 71.7 | 79.6 | 56.1 | 70.2 |
| **AMA-alignment (ours)** | **75.8** | **80.6** | **57.2** | **73.3** |

**Hierarchical Representation Quality** To assess the preservation of semantic hierarchies in learned representations, we employ two metrics: (1) *Hierarchical Clustering Normalized Mutual Information (HC-NMI)* [10], and (2) *Intra-class Variance Reduction (IVR)*. As summarized in Table 2, AMA-alignment achieves superior alignment with ground-truth hierarchies, demonstrating its effectiveness in modeling multi-scale structures.

Table 2: Hierarchical alignment results in DeepFashion. Higher HC-NMI and lower IVR indicate better semantic structure preservation.

| Method | HC-NMI ↑ | IVR ↓ |
|---|---|---|
| SimCLR [7] | 0.52 | 0.134 |
| HardNeg [43] | 0.56 | 0.119 |
| $\alpha$-CL [52] | 0.59 | 0.114 |
| **AMA-alignment (ours)** | **0.66** | **0.091** |

**Ablation Study.** We conduct ablation studies to examine the contributions of the key components in AMA-alignment, as summarized in Table 3. Starting from a global-only contrastive learning (CL) baseline, we progressively introduce multi-scale partitioning and local contrast loss. The results show that both components are essential for achieving robust alignment performance.

Specifically, incorporating the multi-scale local contrastive loss (+ Multi-scale) improves accuracy across all benchmarks, indicating that modeling representations at different semantic granularities might benefit global consistency. In contrast, if we use an adaptive weighting mechanism (+ Adaptive) but implemented via the heuristic loss given in (13), the performance degrades. Unlike the proposed local contrastive loss in (12), this formulation does not restrict negatives to local regions, thereby distorting the affinity structure and degrading performance, as discussed in Remark 1.

In contrast, our AMA-alignment employs the principled local contrastive loss in (12), which rigorously aligns the semantic affinity matrix $K^{sem}$ and the representation affinity matrix $K^{rep}$ through probabilistic graph coupling. This theoretically grounded formulation ensures that both the global and local alignment objectives work coherently in synergy.

Table 3: Ablation study on DeepFashion.

| Accuracy (%) | DeepFashion | iNat | ImageNet |
|---|---|---|---|
| Global-only CL | 70.2 | 54.0 | 69.5 |
| + Multi-scale (fixed weights) | 72.3 | 55.9 | 70.2 |
| + Adaptive (using (13)) | 67.3 | 50.4 | 67.8 |
| **AMA-alignment** | **75.8** | **57.2** | **73.3** |

## 5 Conclusion

We propose a novel hierarchical contrastive learning framework that aligns local affinity structures across multiple scales. By introducing local contrastive objectives and applying a distributionally robust optimization strategy, our method dynamically identifies and refines semantically challenging regions in the representation space. The theoretical analysis establishes convergence guarantees, and extensive experiments showcase its superior performance in capturing hierarchical semantics compared to existing methods. This work advances contrastive methods beyond global alignment, enabling more interpretable and task-specific representations in complex domains.

## Acknowledgments and Disclosure of Funding

This work was supported in part by the National Key R&D program of China through grant 2021YFA1000900, the NSFC through grants No. 62432016 and No. 62272432, and the Provincial NSF of Anhui through grant 2208085MF163. The authors would also like to thank the anonymous reviewers for their valuable comments and suggestions.

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

# A  Other Related Works

There are two key areas of research closely related to our contribution:

**Multi-level Representative Learning**   Several studies have leveraged multi-level labels [25, 66] to learn hierarchical representations, which are supervised learning approaches requiring multi-level labeled data. In contrast, our work focuses on unsupervised learning, aligning with the conventional settings of CL algorithms. Mo et al. [36] proposed to use two types of contrastive losses at shallow and deep network layers, enhancing the quality of learned representations. Additionally, other works [14, 59] embedded representations into hyperbolic space to better model hierarchical relationships. A number of recent studies [32, 54, 12] focused on refining augmentation strategies to better encode semantic information, either directly or indirectly. Our study demonstrates that by more effectively utilizing the semantics encoded in augmentations, we can capture fine-grained semantic structures more accurately through improved optimization techniques.

In the field of manifold learning, Fang and Saad [11] developed a multilevel framework that improves computational efficiency through graph coarsening and multi-scale refinement. Huang et al. [23] introduced a multi-view clustering approach that employs topological manifold learning to better capture the intrinsic nonlinear data relationships. Several works [58, 30, 19, 50, 5, 9] in contrastive clustering have reported high clustering purity. For example, these studies provide various t-SNE or kernel density estimation visualizations to confirm that contrastive objectives naturally facilitate semantic grouping.

**Distributional Robustness**   Distributional robustness [29, 46, 21, 22] addresses the challenge of model performance under distribution shifts by optimizing for worst-case performance across possible distributions. This concept is extended to group distributional robustness optimization (group-DRO) [45], which focuses on maintaining consistent performance across subgroups, including underrepresented ones. Recent work by [53] employs logits adjustment to balance performance across diverse data groups.

# B  Probabilistic Graph Coupling and Its Equivalence to InfoNCE

In this section, we first introduce the foundational concepts of the Probabilistic Graph Coupling Framework, as primarily discussed in Sec. 2.2. We then demonstrate how the InfoNCE loss can be interpreted within this framework. Although the detailed exposition is provided in [55, 50], we include a summary here for completeness and to lay the groundwork for the theoretical analysis in Sec. 3.1.

## B.1  Graph-Based Probabilistic Modeling

Directly comparing and aligning the affinity matrices $\mathbf{K}^{\text{sem}}$ and $\mathbf{K}^{\text{rep}}$ is computationally challenging due to the potentially infinite number of edges in the corresponding graphs. To address this, the probabilistic graph coupling framework [55] reformulates the problem by defining probability distributions over unweighted directed subgraphs. Specifically, it introduces posterior distributions over binary adjacency graphs $\mathbf{W}^{\text{sem}}, \mathbf{W}^{\text{rep}} \in \{0,1\}^{N \times N}$, each sampled according to the affinities $\mathbf{K}^{\text{sem}}$ and $\mathbf{K}^{\text{rep}}$, respectively. The comparison is then framed as minimizing the cross-entropy (7) between these posterior distributions (see (6)), offering a more tractable and principled approach to aligning complex graphs.

**Properties of the Subgraph Probabilistic Distribution** $\Pr(\mathbf{W}|\mathbf{K})$   Consider a graph with an affinity matrix denoted by $\mathbf{K} \in \mathbb{R}_+^{N \times N}$ (either $\mathbf{K}^{\text{sem}}$ or $\mathbf{K}^{\text{rep}}$, as defined in Definition 1 and 2). From $\mathbf{K}$, we sample a subgraph $\mathbf{W}$ (can refer to the aforementioned $\mathbf{W}^{\text{sem}}$ or $\mathbf{W}^{\text{rep}}$) under the topological constraints specified in (5). For clarity, we restate the constraint as follows:

$$\mathcal{S}^{\mathbf{W}} \triangleq \{\mathbf{W} \in \{0,1\}^{N \times N} \mid \forall i \in [N], \sum_j \mathbf{W}_{ij} = 1\},$$

where $\mathbf{W}_{ij} = 1$ indicates the presence of a directed edge from node $i$ to node $j$. This formulation restricts the subgraphs to *binary directed graphs*, where each node has *exactly one* outgoing edge.

Van Assel et al. [55] demonstrates that the posterior probability of sampling a subgraph $\mathbf{W}$ is given by the distribution in (6), namely:

$$\Pr(\mathbf{W}|\mathbf{K}) \propto \Omega(\mathbf{W}) \prod_{(i,j)\in[N]^2} (\mathbf{K}_{ij})^{\mathbf{W}_{ij}},$$

where $\Omega(\mathbf{W}) \triangleq \prod_i \mathbb{1}\left(\sum_j \mathbf{W}_{ij} = 1\right)$ enforces the topological constraint, ensuring that each node has exactly one outgoing edge.

Given the unitary out-degree filter described above, let $\mathbf{W}_i$ and $\mathbf{K}_i$ denote the $i$-th rows of $\mathbf{W}$ and $\mathbf{K}$, respectively. The distribution $\Pr(\mathbf{W}; \mathbf{K})$ satisfies the following property: for $\mathbf{W} \sim \Pr(\cdot; \mathbf{K})$, we have

$$\mathbf{W}_i \sim \text{Multinomial}\left(1, \mathbf{K}_i / \sum_j \mathbf{K}_{ij}\right), \forall i \in [N], \tag{18}$$

where the $\text{Multinomial}(1, \cdot)$ distribution produces a one-hot vector by drawing a single sample according to the normalized weights in $\mathbf{K}_i$. In other words, each node selects exactly one outgoing edge, with probabilities proportional to its affinities. Moreover, the rows $\mathbf{W}_i$ and $\mathbf{W}_{i'}$ are independent for any $i, i' \in [N]$.

## B.2 Connecting Probabilistic Graph Coupling with InfoNCE

According to (3) in Definition 1, the adjacency matrix $\mathbf{K}^{\text{sem}}$ represents the semantic relationships between data points in the augmented data space $X$. For the subgraph $\mathbf{W}^{\text{sem}}$ sampled from the semantic affinity matrix $\mathbf{K}^{\text{sem}}$, based on (18), the probability that there is an edge from $i$ to $j$ is given by:

$$\Pr\left(\mathbf{W}_{ij}^{\text{sem}} = 1 \mid \mathbf{K}^{\text{sem}}\right) \propto \mathbf{K}_{ij}^{\text{sem}} = \mathbb{E}_{\bar{x} \sim P_{\bar{X}}}\left[\Pr_A(x_i|\bar{x}) \Pr_A(x_j|\bar{x})\right], \tag{19}$$

where $\Pr_A(x|\bar{x})$ denotes the augmentation likelihood of $x$ given the natural data point $\bar{x}$.

Now we consider the embedding space. The set of feature embeddings corresponding to the augmented dataset $X$ is represented as $Z \subset \mathcal{Z}$, with each embedding $z_i = g(x_i)$ representing the data point $x_i \in \mathcal{X}$. According to (4) in Definition 2, for any $z_i, z_j \in Z$, the matrix element is defined as:

$$\mathbf{K}_{ij}^{\text{rep}} \triangleq \text{Ker}(z_i, z_j), \text{ where } z_i = g(x_i), z_j = g(x_j).$$

Here $\text{Ker}(\cdot, \cdot)$ is a kernel function. In this work, we use a translation-invariant kernel [55] (*e.g.*, Gaussian kernel).

For the subgraph $\mathbf{W}^{\text{rep}}$ sampled from the representation affinity matrix $\mathbf{K}^{\text{rep}}$, based on (18), the probability that there is an edge from $i$ to $j$ is given by:

$$\Pr\left(\mathbf{W}_{ij}^{\text{rep}} = 1 \mid \mathbf{K}^{\text{rep}}\right) = \frac{\mathbf{K}_{ij}^{\text{rep}}}{\|\mathbf{K}_i^{\text{rep}}\|_1} = \frac{\text{Ker}(z_i, z_j)}{\sum_{k \neq i} \text{Ker}(z_i, z_k)}, \tag{20}$$

The cross-entropy between $\mathbf{W}^{\text{sem}}$ and $\mathbf{W}^{\text{rep}}$ in (7) is decomposed into individual contributions of edges:

$$-\sum_{i=1}^N \sum_{j \neq i}^N \Pr\left(\mathbf{W}_{ij}^{\text{sem}} = 1 \mid \mathbf{K}^{\text{sem}}\right) \log\left(\Pr\left(\mathbf{W}_{ij}^{\text{rep}} = 1 \mid \mathbf{K}^{\text{rep}}\right)\right) \tag{21}$$

$$= -\mathbb{E}_{\bar{x} \sim \mathcal{P}_{\bar{X}}} \mathbb{E}_{x_i, x_j \sim \Pr_A(\cdot|\bar{x})} \left[\log \frac{\text{Ker}(z_i, z_j)}{\sum_{k \neq i} \text{Ker}(z_i, z_k)}\right]. \tag{22}$$

The (22) is obtained by substituting (19) and (20) into (21). If we take the kernel $\text{Ker}(\cdot, \cdot)$ to a Gaussian kernel, namely $\text{Ker}(z_i, z_j) = \exp(-\|z_i - z_j\|^2/\tau)$, we can rewrite the cross-entropy (7) as:

$$-\mathbb{E}_{\bar{x} \sim \mathcal{P}_{\bar{X}}} \mathbb{E}_{x_i, x_j \sim \Pr_A(\cdot|\bar{x})} \left[\log \frac{\exp\left(g(x_i)^\top g(x_j)/\tau\right)}{\sum_{k \neq i} \exp\left(g(x_i)^\top g(x_k)/\tau\right)}\right] + \text{const}, \tag{23}$$

which aligns precisely with the InfoNCE loss (2).

Thus, minimizing (23) matches the InfoNCE objective, thereby establishing a theoretical equivalence between contrastive learning objective and probabilistic graph alignment.

# C Omitted Proofs in Sec. 3.1

In this section, we provide proofs for Proposition 3.1 and Lemma 3.2 in Sec. 3.1. Our goal is to formally establish the connection between (i) the local contrastive objective and the KL divergence over subgraph distributions, and (ii) the subgraph sampling probabilities and row-wise independence, which underpins our localized probabilistic alignment framework.

*Proof of Proposition 3.1.* This result is a direct consequence of the probabilistic graph coupling framework described in Van Assel et al. [55], Tan et al. [50]. Under the constraint $\sum_j \mathbf{W}_{ij} = 1$ for all $i$, each row of $\mathbf{W}$ defines a multinomial distribution over neighbors. We use $\mathbf{W}_{i\cdot}$ to denote the $i$-th row of matrix $\mathbf{W}$, then the full sampling distribution factorizes row-wise:

$$\Pr(\mathbf{W} \mid \mathbf{K}_s) = \prod_{i=1}^{N'} \Pr(\mathbf{W}_{i\cdot} \mid \mathbf{K}_{s,i\cdot}) = \prod_{i=1}^{N'} \mathrm{Multinomial}\Big(1; \mathbf{K}_{s,i\cdot}/\sum\nolimits_j \mathbf{K}_{s,ij}\Big), \qquad (24)$$

which corresponds to drawing one outgoing edge from each node independently, proportional to the row-wise kernel similarity. $\qquad\square$

**Lemma C.1** (Restatement of Lemma 3.2). *Let $\mathbf{W}_s^{\mathtt{sem}}$ and $\mathbf{W}_s^{\mathtt{rep}}$ be subgraphs sampled from $\mathbf{K}_s^{\mathtt{sem}}$ and $\mathbf{K}_s^{\mathtt{rep}}$, respectively, subject to the topological constraints in (5). Minimizing the cross-entropy between the distributions of these subgraphs, i.e.,*

$$\min_{g(\cdot)} \ -\mathbb{E}_{\mathbf{W}_s^{\mathtt{sem}} \sim \Pr(\cdot \mid \mathbf{K}_s^{\mathtt{sem}})}\Big[\log \Pr(\mathbf{W}_s^{\mathtt{rep}} = \mathbf{W}_s^{\mathtt{sem}} \mid \mathbf{K}_s^{\mathtt{rep}})\Big] \qquad (25)$$

*is equivalent to minimizing the local contrastive loss $\mathcal{L}^s$ defined in (12).*

*Proof.* Recall that the local region $R_r^s(\theta, z_{\mathrm{anc}}) = \{z \in \mathbb{S}^{d_g-1} : z^\top z_{\mathrm{anc}} \leq r_s\}$ selects points whose representations lie within a hyperspherical cap centered at $z_{\mathrm{anc}}$. Let $I_{R_r^s}$ denote the indices of points in $X$ falling within region $R_r^s$. Then the local semantic affinity matrix $\mathbf{K}_s^{\mathtt{sem}}$ is the submatrix of $\mathbf{K}^{\mathtt{sem}}$ restricted to $I_{R_r^s} \times I_{R_r^s}$, and similarly for the representation affinity matrix $\mathbf{K}_s^{\mathtt{rep}}$.

We sample subgraphs $\mathbf{W}_s^{\mathtt{sem}}$ from $\mathbf{K}_s^{\mathtt{sem}}$ and $\mathbf{W}_s^{\mathtt{rep}}$ from $\mathbf{K}_s^{\mathtt{rep}}$, each subject to the topological constraints:

$$\mathbf{W}_s^{\mathtt{sem}} \in \mathcal{S}^{\mathbf{W}_s^{\mathtt{sem}}}, \quad \mathbf{W}_s^{\mathtt{rep}} \in \mathcal{S}^{\mathbf{W}_s^{\mathtt{rep}}}, \qquad (26)$$

$$\text{where} \quad \mathcal{S}^{\mathbf{W}_s} \triangleq \Big\{\mathbf{W}_s \in \{0,1\}^{N' \times N'} \big| \sum_{j=1}^{N'} \mathbf{W}_{s,ij} = 1, \forall i\Big\} \quad \text{for } \mathbf{W}_s \in \{\mathbf{W}_s^{\mathtt{sem}}, \mathbf{W}_s^{\mathtt{rep}}\}, \qquad (27)$$

where $\mathcal{S}^{\mathbf{W}_s}$ denotes the set of admissible sampled graphs from $\mathbf{K}_s \in \{\mathbf{K}_s^{\mathtt{sem}}, \mathbf{K}_s^{\mathtt{rep}}\}$. That is, each node $i$ has exactly one outgoing edge in both sampled graphs $\mathbf{W}_s^{\mathtt{sem}}$ and $\mathbf{W}_s^{\mathtt{rep}}$ respectively.

By Proposition 3.1, the probability of subgraph $\mathbf{W}_s^{\mathtt{sem}}$ given $\mathbf{K}_s^{\mathtt{sem}}$ factorizes row-by-row:

$$\Pr(\mathbf{W}_s^{\mathtt{sem}} \mid \mathbf{K}_s^{\mathtt{sem}}) \propto \prod_{i=1}^{N'} \prod_{j=1}^{N'} \Big(\mathbf{K}_{s,ij}^{\mathtt{sem}}\Big)^{\mathbf{W}_{s,ij}^{\mathtt{sem}}}, \quad \text{with the constraint } \sum_{j=1}^{N'} \mathbf{W}_{s,ij}^{\mathtt{sem}} = 1, \ \forall i. \qquad (28)$$

Specifically, $\mathbf{W}_{s,ij}^{\mathtt{sem}} = 1$ means node $i$ connects to node $j$. The same holds for $\mathbf{W}_s^{\mathtt{rep}} \sim \Pr(\cdot \mid \mathbf{K}_s^{\mathtt{rep}})$.

Using the factorized form (28) for both subgraph distributions, we decompose the cross-entropy (7) sum over all $i$ (rows) and $j$ (possible targets of node $i$),

$$-\sum_{i=1}^{N} \sum_{j \neq i}^{N} \Pr\big(\mathbf{W}_{s,ij}^{\mathtt{sem}} = 1 \mid \mathbf{K}_s^{\mathtt{sem}}\big) \log \big(\Pr\big(\mathbf{W}_{s,ij}^{\mathtt{rep}} = 1 \mid \mathbf{K}_s^{\mathtt{rep}}\big)\big). \qquad (29)$$

According to Definition 1, we have $\Pr\big(\mathbf{W}_{ij}^{\mathtt{sem}} = 1 \mid \mathbf{K}^{\mathtt{sem}}\big) = \mathbb{E}_{\bar{x} \sim P_{\bar{X}}}\left[\Pr_{\mathrm{A}}(x_i|\bar{x}) \cdot \Pr_{\mathrm{A}}(x_j|\bar{x})\right]$. By the definition of $\Pr_r^s$ in (11), we have

$$\Pr\big(\mathbf{W}_{s,ij}^{\mathtt{sem}} = 1 \mid \mathbf{K}_s^{\mathtt{sem}}\big) \propto \mathbb{E}_{\bar{x} \sim P_{\bar{X}}}\left[\Pr_r^s(x_i|\bar{x}) \cdot \Pr_r^s(x_j|\bar{x})\right]. \qquad (30)$$

Further, the probability that the sampled graph $\mathbf{W}_{s,ij}^{\texttt{rep}}$ has node $i$ connecting to node $j$ under $\mathbf{K}_s^{\texttt{rep}}$ is given by

$$\Pr\left(\mathbf{W}_{s,ij}^{\texttt{rep}} = 1 \;\middle|\; \mathbf{K}_s^{\texttt{rep}}\right) = \frac{\mathbf{K}_{s,ij}^{\texttt{rep}}}{\sum_{k \neq i} \mathbb{1}_{g(x_i),g(x_k) \in R_r^s}\left(\mathbf{K}_{s,ik}^{\texttt{rep}}\right)}. \tag{31}$$

Similar with Appendix B.2, if we take the kernel $\texttt{Ker}(\cdot,\cdot)$ to a Gaussian kernel, namely $\texttt{Ker}(z_i, z_j) = \exp(-\|z_i - z_j\|^2/\tau)$, we can rewrite the above probability (31) as

$$\Pr\left(\mathbf{W}_{s,ij}^{\texttt{rep}} = 1 \mid \mathbf{K}_s^{\texttt{rep}}\right) \;=\; \frac{\exp\left(z_i^\top z_j/\tau\right)}{\sum_{k \neq i} \mathbb{1}_{g(x_i),g(x_k) \in R_r^s}\left(\exp(z_i^\top z_k/\tau)\right)}, \tag{32}$$

where $\mathbb{1}_{g(x_i),g(x_k) \in R_r^s}$ is the indicator function ensures that $g(x_i)$ and $g(x_k)$ lie in $R_r^s$. Finally, we can derive the cross-entropy by inserting (30) and (32) into (29),

$$-\mathop{\mathbb{E}}_{\bar{x} \sim \mathcal{P}_{\bar{X}}} \mathop{\mathbb{E}}_{x_i, x_j \sim \Pr_r^s(\cdot|\bar{x})} \left[\log \frac{\exp\left(z_i^\top z_j/\tau\right)}{\sum_{k \neq i} \mathbb{1}_r^s(x_k)\left(\exp(z_i^\top z_k/\tau)\right)}\right] + \text{const},$$

which exactly matches the local contrastive loss $\mathcal{L}^s$ in (12) up to a constant. Thus, minimizing the cross-entropy between local subgraph distributions is equivalent to minimizing the local contrastive loss. $\qquad\square$

# D   Theoretical Analysis for AMA-alignment

In this section, we establish the convergence rate of our proposed method: AMA-alignment (Algorithm 1).

**Overview of the analysis.**    We begin by formalizing the multi-scale min–max alignment objective and deriving the closed-form solution for the inner weight maximization. Based on this formulation, we justify the update rules for both the weight vector $\mathbf{q}$ and the model parameters $\theta$ used in Algorithm 1 (Sec. D.1). Next, we introduce the boundedness and smoothness assumptions required for the theoretical analysis. We prove that the optimal weight $\mathbf{q}^*$ is Lipschitz continuous w.r.t. $\theta$. Based on these, we further show that the overall objective function is smooth (Sec. D.2). We then analyze the recursive exponentiated gradient updates (17), demonstrating that they track the optimal weights with an $O(1/t)$ error rate under stochastic updates. Finally, by quantifying the gradient approximation error and applying a single-step descent lemma, we telescope the inequalities to establish the $O(1/\sqrt{T})$ convergence rate (Sec. D.4).

## D.1   Problem Formulation and Basic Properties

In this section, we introduce the overall min–max structure of our adaptive multi-scale alignment objective. We first formalize the outer minimization over the encoder parameters $\theta$ and the inner maximization over region-weight distributions $\mathbf{q}^s$ (35). The algorithmic procedures are based on two technical results: (i) the closed-form solution of the inner maximization (Proposition D.1) and (ii) the gradient formula for the AMA-alignment objective (34) (Lemma D.2).

### D.1.1   Restate of Algorithm 1 and Setup for Analysis

We study the following multi-scale min–max formulation:

$$\min_{\theta \in \Theta} \sum_{s=1}^S \max_{\mathbf{q}^s \in \Delta_{m^s}} \left\{ \sum_{r=1}^{m^s} \mathbf{q}_r^s \, \mathcal{L}^s(\theta, R_r^s) - \rho \, \mathbf{KL}(\mathbf{q}^s \| \mathbf{u}) \right\}, \tag{33}$$

where $\mathcal{L}^s(\theta, R_r^s)$ denotes the local contrastive loss at region $R_r^s$, and the KL term regularizes the weight distribution $\mathbf{q}^s$ against the uniform distribution $\mathbf{u}$. The full objective is

$$F(\theta) = \sum_{s=1}^S F^s(\theta), \tag{34}$$

where for each scale $s$, the scale-specific objective is

$$F^s(\theta) = \max_{\mathbf{q}^s \in \Delta_{m^s}} h^s(\theta, \mathbf{q}^s), \quad h^s(\theta, \mathbf{q}^s) := \sum_{r=1}^{m^s} \mathbf{q}_r^s \mathcal{L}^s(\theta, R_r^s) - \rho \mathbf{KL}(\mathbf{q}^s \| \mathbf{u}). \tag{35}$$

Although the inner maximizer $\mathbf{q}^{s,*}(\theta)$ in (35) admits a closed-form expression, this optimal solution depends on the current encoder parameters $\theta$, which vary across iterations. Furthermore, in a stochastic setting, we only observe noisy estimates of the true loss vector (we denote $\hat{\mathcal{L}}^s(\theta_t, R_r^s)$ the estimate of $\mathcal{L}^s(\theta_t, R_r^s)$). A naive update that directly computes the closed-form solution based on a noisy loss estimate for a moving target $\theta_t$ would be unstable.

For both stability and efficiency, we therefore adopt a stochastic mirror ascent scheme. This approach smoothly tracks the moving optimum by regularizing the update step, keeping the new weight distribution $\mathbf{q}_{t+1}^s$ close to the previous one $\mathbf{q}_t^s$. This leads to the following regularized optimization problem for updating the weights from iteration $t$ to $t+1$:

$$\mathbf{q}_{t+1}^s \in \operatorname*{argmax}_{\mathbf{q}^s \in \Delta_{m^s}} \left\{ \sum_r \mathbf{q}_r^s \hat{\mathcal{L}}^s(\theta_{t+1}, R_r^s) - \rho \mathbf{KL}(\mathbf{q}^s \| \mathbf{u}) - \frac{1}{\eta_{\mathbf{q},t}} \mathbf{KL}(\mathbf{q}^s \| \mathbf{q}_t^s) \right\}. \tag{36}$$

This step both exploits the new loss signal from the updated parameters $\theta_{t+1}$ and ensures a stable evolution of the weights.

**The update rules in Algorithm 1 are summarized as follows.**
*Updating Encoder Parameters ($\theta$):* The encoder is updated via stochastic gradient descent using the weights from the beginning of the iteration.

$$\theta_{t+1} = \theta_t - \eta_{\theta,t}\, \mathbf{g}_t, \quad \text{where} \quad \mathbf{g}_t = \sum_{s=1}^{S} \sum_{r=1}^{m^s} \mathbf{q}_{t,r}^s \nabla \hat{\mathcal{L}}^s(\theta_t, R_r^s). \tag{37}$$

*Updating Region Weights ($\mathbf{q}^s$):* After updating $\theta_t$ to $\theta_{t+1}$, the region weights are updated by solving the mirror ascent problem (36). As established in Proposition D.1, this yields the closed-form solution:

$$\mathbf{q}_{t+1,r}^s = \frac{\left(\mathbf{q}_{t,r}^s\right)^{1/\gamma_t} \exp\left(\frac{\eta_{\mathbf{q},t}}{\gamma_t} \hat{\mathcal{L}}^s(\theta_{t+1}, R_r^s)\right)}{\sum_{j=1}^{m^s} \left(\mathbf{q}_{t,j}^s\right)^{1/\gamma_t} \exp\left(\frac{\eta_{\mathbf{q},t}}{\gamma_t} \hat{\mathcal{L}}^s(\theta_{t+1}, R_j^s)\right)}, \tag{38}$$

where $\gamma_t = 1 + \rho\, \eta_{\mathbf{q},t}$. These two updates for $\theta$ and $\{\mathbf{q}^s\}$ alternate until convergence.

**Proposition D.1** (Closed-form solution for regularized mirror ascent update). *For the update step from iteration $t$ to $t+1$, the optimal solution $\mathbf{q}_{t+1}^s$ to the regularized mirror ascent problem defined in (36) is given by (38). Moreover, for any $\rho > 0$ and $\eta_{\mathbf{q},t} > 0$, this solution is unique and strictly positive (i.e., $\mathbf{q}_{t+1}^s \in \operatorname{interior}(\Delta_{m^s})$).*

*Proof.* The objective in (36) is strictly concave in $\mathbf{q}^s$ because both $-\rho\, \mathrm{KL}(\mathbf{q}^s \| \mathbf{u})$ and $-\frac{1}{\eta_{\mathbf{q},t}} \mathrm{KL}(\mathbf{q}^s \| \mathbf{q}_t^s)$ are strictly concave in $\mathbf{q}^s$ on the simplex, and the remaining term is linear; hence the maximizer is unique.

To find this solution, we form the Lagrangian, incorporating the simplex constraint $\sum_{r=1}^{m^s} \mathbf{q}_r^s = 1$:

$$\text{Lagrangian}(\mathbf{q}^s, \lambda) = \sum_{r=1}^{m^s} \mathbf{q}_r^s \hat{\mathcal{L}}^s(\theta_{t+1}, R_r^s) - \rho \sum_{r=1}^{m^s} \mathbf{q}_r^s \log(m^s \mathbf{q}_r^s)$$

$$- \frac{1}{\eta_{\mathbf{q},t}} \sum_{r=1}^{m^s} \mathbf{q}_r^s \log\left(\frac{\mathbf{q}_r^s}{\mathbf{q}_{t,r}^s}\right) + \lambda\left(1 - \sum_{r=1}^{m^s} \mathbf{q}_r^s\right).$$

Setting the derivative with respect to $\mathbf{q}_r^s$ to zero yields the KKT condition:

$$\hat{\mathcal{L}}^s(\theta_{t+1}, R_r^s) - \rho(\log(m^s \mathbf{q}_r^s) + 1) - \frac{1}{\eta_{\mathbf{q},t}}\left(\log\left(\frac{\mathbf{q}_r^s}{\mathbf{q}_{t,r}^s}\right) + 1\right) - \lambda = 0.$$

We collect terms involving $\log(\mathbf{q}_r^s)$ and rearrange:

$$\left(\rho + \tfrac{1}{\eta_{\mathbf{q},t}}\right)\log(\mathbf{q}_r^s) = \hat{\mathcal{L}}^s(\theta_{t+1}, R_r^s) + \tfrac{1}{\eta_{\mathbf{q},t}}\log(\mathbf{q}_{t,r}^s) - \rho\log(m^s) - C(\lambda),$$

where $C(\lambda) = \rho + 1/\eta_{\mathbf{q},t} + \lambda$ groups all terms independent of the index $r$. Let $\gamma_t = 1 + \rho\eta_{\mathbf{q},t}$. The term on the left is $(\gamma_t/\eta_{\mathbf{q},t})$. Multiplying by $\eta_{\mathbf{q},t}/\gamma_t$ isolates $\log(\mathbf{q}_r^s)$:

$$\log(\mathbf{q}_r^s) = \tfrac{\eta_{\mathbf{q},t}}{\gamma_t}\hat{\mathcal{L}}^s(\theta_{t+1}, R_r^s) + \tfrac{1}{\gamma_t}\log(\mathbf{q}_{t,r}^s) - C'(\lambda),$$

where $C'(\lambda)$ is another constant absorbing $\lambda$, $\rho$, and $m^s$. Exponentiating both sides gives

$$\mathbf{q}_r^s = \exp(-C'(\lambda)) \cdot (\mathbf{q}_{t,r}^s)^{1/\gamma_t} \cdot \exp\left(\tfrac{\eta_{\mathbf{q},t}}{\gamma_t}\hat{\mathcal{L}}^s(\theta_{t+1}, R_r^s)\right).$$

The term $\exp(-C'(\lambda))$ acts as a normalization constant, which is determined by the constraint $\sum_{k=1}^{m^s}\mathbf{q}_k^s = 1$. Substituting the expression for $\mathbf{q}_r^s$ into this constraint and solving for the constant immediately yields the closed form in (38) for $\mathbf{q}_{t+1,r}^s$.

Since $\rho > 0$, $\eta_{\mathbf{q},t} > 0$, and by induction $\mathbf{q}_{t,r}^s > 0$, the numerator of (38) is strictly positive for all $r$. This ensures that the solution $\mathbf{q}_{t+1}^s$ is unique and lies in the interior of the simplex. By strict concavity and Slater's condition, the KKT conditions are necessary and sufficient, completing the proof. $\square$

Next, we justify the form of the outer update in (16) as the following lemma.

**Lemma D.2** (Gradient of the Objective Function). *If $\mathcal{L}^s(\theta, R_r^s)$ is continuously differentiable with respect to $\theta$ for each region $r$, and $\rho > 0$, then $F^s(\theta)$ (35) is differentiable with $\nabla_\theta F^s(\theta) = \sum_{r=1}^{m^s}\mathbf{q}_r^{s,*}(\theta)\nabla_\theta\mathcal{L}^s(\theta, R_r^s)$ and $F(\theta)$ (34) is differentiable and its gradient is given by:*

$$\nabla_\theta F(\theta) = \sum_{s=1}^{S}\nabla_\theta F^s(\theta) = \sum_{s=1}^{S}\sum_{r=1}^{m^s}\mathbf{q}_r^{s,*}(\theta)\nabla_\theta\mathcal{L}^s(\theta, R_r^s) \tag{39}$$

*where $\mathbf{q}^{s,*}(\theta)$ is the unique optimal distribution characterized in Proposition D.1.*

*Proof.* The result follows from a direct application of Danskin's theorem for max-functions with regularization. We verify that all necessary conditions are satisfied:

*(i) Compactness of the constraint set:* The probability simplex $\Delta_{m^s}$ is compact.

*(ii) Continuity and differentiability:* The objective function $h^s(\theta, \mathbf{q}^s)$ defined in (35) is continuous in the pair $(\theta, \mathbf{q})$ and continuously differentiable in $\theta$ for each fixed $\mathbf{q}$, as $\mathcal{L}^s(\theta, R_r^s)$ is continuously differentiable in $\theta$ by assumption.

*(iii) Uniqueness of maximizer:* As established in Proposition D.1, for each fixed $\theta$, the KL-regularization term ensures that the maximizer $\mathbf{q}^{s,*}(\theta)$ is unique and lies in the interior of $\Delta_{m^s}$.

Since all conditions are satisfied, Danskin's theorem implies that $F^s(\theta)$ is differentiable and its gradient equals the partial gradient of $h^s$ (defined in (35)) with respect to $\theta$ evaluated at the optimal point:

$$\nabla_\theta F^s(\theta) = \nabla_\theta h^s(\theta, \mathbf{q}^{s,*}(\theta))$$

$$= \nabla_\theta\left(\sum_{r=1}^{m^s}\mathbf{q}_r^{s,*}(\theta)\mathcal{L}^s(\theta, R_r^s) - \rho\mathbf{KL}(\mathbf{q}^{s,*}(\theta)\|\mathbf{u})\right)$$

$$= \sum_{r=1}^{m^s}\mathbf{q}_r^{s,*}(\theta)\nabla_\theta\mathcal{L}^s(\theta, R_r^s) \tag{40}$$

The KL-regularization term vanishes in the gradient computation because it depends on $\theta$ only through $\mathbf{q}^{s,*}(\theta)$, and the envelope theorem accounts for these implicit dependencies.

This elegant formula enables efficient gradient computation without requiring the derivative of $\mathbf{q}^{s,*}(\theta)$ with respect to $\theta$, which would be computationally intensive. For the full model with multiple scenarios, the gradient of the overall objective $F(\theta) = \sum_{s=1}^{S}F^s(\theta)$ (34) is simply:

$$\nabla F(\theta) = \sum_{s=1}^{S}\nabla F^s(\theta) = \sum_{s=1}^{S}\sum_{r=1}^{m^s}\mathbf{q}_r^{s,*}(\theta)\nabla_\theta\mathcal{L}^s(\theta, R_r^s)$$

$\square$

## D.2 Key Assumptions and Properties of the Adaptive Weight Model

We now state the technical assumptions underlying our convergence analysis, which include boundedness and smoothness conditions (Assumptions 1–4). Under these assumptions, we establish two fundamental properties: (i) the Lipschitz continuity of the optimal weight mapping $\theta \mapsto \mathbf{q}^{s,*}(\theta)$ (Lemma D.3), and (ii) the smoothness of the composite objective $F(\theta)$ (Lemma D.4). Together, these results guarantee that the interaction between the inner maximization and the outer minimization remains stable and analytically tractable. Unless otherwise specified, all vector norms are Euclidean and all matrix norms are spectral.

**Assumption 1** (Gradient Boundedness). For all $\theta \in \Theta$, scales $s$, and regions $R_r^s$, $\|\nabla_\theta \mathcal{L}^s(\theta, R_r^s)\| \leq G$.

**Assumption 2** (Loss Boundedness). For all $\theta \in \Theta$, regions $R_r^s$, and scales $s$, there exists a constant $L_{\max}$ such that $0 \leq \mathcal{L}^s(\theta, R_r^s) \leq L_{\max}$.

**Assumption 3** (Smoothness). For each scale $s$ and region $r$, $\mathcal{L}^s(\theta, R_r^s)$ is $L$-smooth in $\theta$: $\|\nabla_\theta \mathcal{L}^s(\theta_1, R_r^s) - \nabla_\theta \mathcal{L}^s(\theta_2, R_r^s)\| \leq L\|\theta_1 - \theta_2\|$ for all $\theta_1, \theta_2 \in \Theta$.

**Assumption 4** (Two-Batch Sampling). At each iteration $t$, we use two independent mini-batches: one for computing the region loss estimates to update the weights $\mathbf{q}^s(\theta_t)$, and another for computing unbiased stochastic gradients $\nabla \hat{\mathcal{L}}^s(\theta_t, R_r^s)$. For constants $\sigma_{\mathcal{L}}^2$ and $\sigma_g^2$, the stochastic loss satisfy:

$$\mathbb{E}[\hat{\mathcal{L}}^s(\theta_t, R_r^s)] = \mathcal{L}^s(\theta_t, R_r^s); \quad \mathbb{E}[|\hat{\mathcal{L}}^s(\theta_t, R_r^s) - \mathcal{L}^s(\theta_t, R_r^s)|] \leq \sigma_{\mathcal{L}}; \tag{41}$$

the stochastic gradients satisfy:

$$\mathbb{E}[\nabla \hat{\mathcal{L}}^s(\theta_t, R_r^s)] = \nabla \mathcal{L}^s(\theta_t, R_r^s); \quad \mathbb{E}[\|\nabla \hat{\mathcal{L}}^s(\theta_t, R_r^s) - \nabla \mathcal{L}^s(\theta_t, R_r^s)\|^2] \leq \sigma_g^2. \tag{42}$$

**Remark 2.** In practice, a single minibatch is often reused for both the weight update and the gradient step. Although this induces a mild statistical dependence between $\mathbf{q}^s(\theta_t)$ and the gradient estimator, our experiments show no degradation in convergence. For theoretical rigor, we adhere to the two-batch model in our theoretical development.

**Lemma D.3** (Lipschitz Continuity of Optimal Weights). *Under Assumptions 1–3, the map $\theta \mapsto \mathbf{q}^{s,*}(\theta)$ that gives the unique solution to the inner maximization in (35) is Lipschitz continuous. Specifically, for any $\theta_1, \theta_2 \in \Theta$:*

$$\|\mathbf{q}^{s,*}(\theta_1) - \mathbf{q}^{s,*}(\theta_2)\| \leq \frac{G\sqrt{m^s}}{\rho}\|\theta_1 - \theta_2\|. \tag{43}$$

*Proof.* The function $h^s(\theta, \mathbf{q}^s)$ defined in (35) is $\rho$-strongly concave with respect to $\mathbf{q}^s$ due to the KL-divergence term. Thus, for any fixed $\theta$, there exists a unique maximizer $\mathbf{q}^{s,*}(\theta)$. This mapping is differentiable, and we can analyze its Jacobian $\frac{\partial \mathbf{q}^{s,*}}{\partial \theta}$ by applying the implicit function theorem to the Karush-Kuhn-Tucker (KKT) conditions.

The KKT condition for the inner maximization problem is $\nabla_{\mathbf{q}} h^s(\theta, \mathbf{q}^{s,*}(\theta)) + \lambda \mathbf{1} = \mathbf{0}$, where $\lambda$ is the Lagrange multiplier for the constraint $\sum_r \mathbf{q}_r^s = 1$. Differentiating this system with respect to $\theta$ yields:

$$\nabla_{\theta\mathbf{q}}^2 h^s(\theta, \mathbf{q}^{s,*}) + \nabla_{\mathbf{q}\mathbf{q}}^2 h^s(\theta, \mathbf{q}^{s,*}) \cdot \frac{\partial \mathbf{q}^{s,*}}{\partial \theta} + \frac{\partial \lambda}{\partial \theta}\mathbf{1} = \mathbf{0}. \tag{44}$$

Let $P = I - \frac{1}{m^s}\mathbf{1}\mathbf{1}^T$ be the projection matrix onto the tangent space of the simplex, $\{v \in \mathbb{R}^{m^s} | \mathbf{1}^T v = 0\}$. Since $\mathbf{1}^T \mathbf{q}^{s,*}(\theta) = 1$ for all $\theta$, its derivative $\frac{\partial \mathbf{q}^{s,*}}{\partial \theta}$ lies in this tangent space, meaning $P\frac{\partial \mathbf{q}^{s,*}}{\partial \theta} = \frac{\partial \mathbf{q}^{s,*}}{\partial \theta}$. Projecting (44) with $P$ eliminates the Lagrange multiplier term (as $P\mathbf{1} = \mathbf{0}$):

$$P\nabla_{\theta\mathbf{q}}^2 h^s(\theta, \mathbf{q}^{s,*}) + P\nabla_{\mathbf{q}\mathbf{q}}^2 h^s(\theta, \mathbf{q}^{s,*})\frac{\partial \mathbf{q}^{s,*}}{\partial \theta} = \mathbf{0}.$$

Since $\frac{\partial \mathbf{q}^{s,*}}{\partial \theta}$ is already in the tangent space, we can write $P\nabla_{\mathbf{q}\mathbf{q}}^2 h^s P\frac{\partial \mathbf{q}^{s,*}}{\partial \theta}$ and rearrange to solve for the Jacobian:

$$\frac{\partial \mathbf{q}^{s,*}}{\partial \theta} = -\left(P\nabla_{\mathbf{q}\mathbf{q}}^2 h^s P\right)^\dagger P\nabla_{\theta\mathbf{q}}^2 h^s,$$

where $(\cdot)^\dagger$ denotes the pseudoinverse, which acts as the inverse on the tangent space.

We now bound the norms of the two matrix terms.

**(1) Hessian Term:** The Hessian of $h^s$ with respect to $\mathbf{q}^s$ is $\nabla^2_{\mathbf{qq}} h^s = -\rho \operatorname{diag}(1/\mathbf{q}_1^{s,*}, \ldots, 1/\mathbf{q}_{m^s}^{s,*})$. Due to $\rho$-strong concavity, the projected negative Hessian is positive definite on the tangent space with its smallest eigenvalue being at least $\rho$. Thus, the operator norm of its inverse on this subspace is bounded:

$$\left\| \left( P \nabla^2_{\mathbf{qq}} h^s P \right)^\dagger \right\| \le \frac{1}{\rho}.$$

**(2) Mixed-Derivative Term:** The mixed derivative is the Jacobian matrix whose $r$-th row is $\nabla_\theta \mathcal{L}^s(\theta, R_r^s)$. We can bound its Frobenius norm using Assumption 1:

$$\|\nabla^2_{\theta \mathbf{q}} h^s\| \le \|\nabla^2_{\theta \mathbf{q}} h^s\|_F = \left( \sum_{r=1}^{m^s} \|\nabla_\theta \mathcal{L}^s(\theta, R_r^s)\|^2 \right)^{1/2} \le \left( \sum_{r=1}^{m^s} G^2 \right)^{1/2} = G\sqrt{m^s}.$$

Combining these bounds and using $\|P\| \le 1$, we get:

$$\left\| \frac{\partial \mathbf{q}^{s,*}}{\partial \theta} \right\| \le \left\| \left( P \nabla^2_{\mathbf{qq}} h^s P \right)^\dagger \right\| \cdot \|P\| \cdot \|\nabla^2_{\theta \mathbf{q}} h^s\| \le \frac{1}{\rho} \cdot 1 \cdot G\sqrt{m^s} = \frac{G\sqrt{m^s}}{\rho}.$$

By the Mean Value Theorem, for any $\theta_1, \theta_2 \in \Theta$, there exists some $\tilde{\theta}$ on the line segment between them such that:

$$\|\mathbf{q}^{s,*}(\theta_1) - \mathbf{q}^{s,*}(\theta_2)\| \le \left\| \frac{\partial \mathbf{q}^{s,*}}{\partial \theta} \bigg|_{\theta=\tilde{\theta}} \right\| \|\theta_1 - \theta_2\| \le \frac{G\sqrt{m^s}}{\rho} \|\theta_1 - \theta_2\|.$$

This establishes the Lipschitz continuity of $\mathbf{q}^{s,*}(\theta)$ with constant $L_q = \frac{G\sqrt{m^s}}{\rho}$. $\qquad \square$

**Lemma D.4** (Smoothness of $F(\theta)$ in (34))**.** *Under Assumptions 1–3, let $M = \max_s m^s$. Each function $F^s(\theta)$ is $L_{F^s}$-smooth with a constant $L_{F^s} \le L + \frac{MG^2}{\rho}$. The overall objective $F(\theta) = \sum_{s=1}^S F^s(\theta)$ is $L_F$-smooth with $L_F \le S(L + \frac{MG^2}{\rho})$.*

*Proof.* For any $\theta_1, \theta_2 \in \Theta$, by Lemma D.2, we have:

$$\|\nabla_\theta F^s(\theta_1) - \nabla_\theta F^s(\theta_2)\| = \Big\| \sum_{r=1}^{m^s} \mathbf{q}_r^{s,*}(\theta_1) \nabla_\theta \mathcal{L}^s(\theta_1, R_r^s) - \sum_{r=1}^{m^s} \mathbf{q}_r^{s,*}(\theta_2) \nabla_\theta \mathcal{L}^s(\theta_2, R_r^s) \Big\|$$

$$= \Big\| \sum_{r=1}^{m^s} \mathbf{q}_r^{s,*}(\theta_1) \nabla_\theta \mathcal{L}^s(\theta_1, R_r^s) - \sum_{r=1}^{m^s} \mathbf{q}_r^{s,*}(\theta_1) \nabla_\theta \mathcal{L}^s(\theta_2, R_r^s)$$

$$+ \sum_{r=1}^{m^s} \mathbf{q}_r^{s,*}(\theta_1) \nabla_\theta \mathcal{L}^s(\theta_2, R_r^s) - \sum_{r=1}^{m^s} \mathbf{q}_r^{s,*}(\theta_2) \nabla_\theta \mathcal{L}^s(\theta_2, R_r^s) \Big\|$$

$$\le \Big\| \sum_{r=1}^{m^s} \mathbf{q}_r^{s,*}(\theta_1) (\nabla_\theta \mathcal{L}^s(\theta_1, R_r^s) - \nabla_\theta \mathcal{L}^s(\theta_2, R_r^s)) \Big\|$$

$$+ \Big\| \sum_{r=1}^{m^s} (\mathbf{q}_r^{s,*}(\theta_1) - \mathbf{q}_r^{s,*}(\theta_2)) \nabla_\theta \mathcal{L}^s(\theta_2, R_r^s) \Big\| \tag{45}$$

For the first term, by Assumption 3 and the fact that $\sum_{r=1}^{m^s} \mathbf{q}_r^{s,*}(\theta_1) = 1$:

$$\Big\| \sum_{r=1}^{m^s} \mathbf{q}_r^{s,*}(\theta_1) (\nabla_\theta \mathcal{L}^s(\theta_1, R_r^s) - \nabla_\theta \mathcal{L}^s(\theta_2, R_r^s)) \Big\| \le \sum_{r=1}^{m^s} \mathbf{q}_r^{s,*}(\theta_1) L \|\theta_1 - \theta_2\| = L \|\theta_1 - \theta_2\| \tag{46}$$

For the second term, by Lemma D.3 and Assumption 1:

$$\sum_{r=1}^{m^s} |\mathbf{q}_r^{s,*}(\theta_1) - \mathbf{q}_r^{s,*}(\theta_2)| \cdot \|\nabla_\theta \mathcal{L}^s(\theta_2, R_r^s)\| \le G \cdot \|\mathbf{q}^{s,*}(\theta_1) - \mathbf{q}^{s,*}(\theta_2)\|_1$$

$$\le G \cdot \sqrt{m^s}\|\mathbf{q}^{s,*}(\theta_1) - \mathbf{q}^{s,*}(\theta_2)\|_2 \le G \cdot \sqrt{m^s}\left(\frac{G\sqrt{m^s}}{\rho}\|\theta_1 - \theta_2\|\right)$$

$$= \frac{m^s G^2}{\rho}\|\theta_1 - \theta_2\|.$$

Combining the two bounds and using $m^s \le M$:

$$\|\nabla_\theta F^s(\theta_1) - \nabla_\theta F^s(\theta_2)\| \le \left(L + \frac{m^s G^2}{\rho}\right)\|\theta_1 - \theta_2\| \le \left(L + \frac{MG^2}{\rho}\right)\|\theta_1 - \theta_2\|.$$

This establishes that each $F^s(\theta)$ is smooth with a constant $L_{F^s} \le L + \frac{MG^2}{\rho}$. Since $F(\theta) = \sum_{s=1}^{S} F^s(\theta)$, the smoothness of the sum is the sum of the smoothness constants (by triangle inequality), so $F(\theta)$ is $L_F$-smooth with $L_F = \sum_s L_{F^s} \le S(L + \frac{MG^2}{\rho})$. $\qquad\square$

### D.3 Analysis of Stochastic Weight Updates

We analyze the recursive stochastic exponentiated–gradient (EG) updates for the region–weight distributions $\mathbf{q}^s$ defined in (36). Our goal is to show that, despite stochastic noise and the slow drift of the encoder parameters $\theta_t$, the iterates $\mathbf{q}_t^s$ track the instantaneous maximizer $\mathbf{q}^{s,*}(\theta_t)$ with vanishing error. In particular, we establish an $O(1/t)$ rate for the expected tracking error $\mathbb{E}[\mathbf{KL}(\mathbf{q}^{s,*}(\theta_t)\|\mathbf{q}_t^s)]$.

For simplicity we fix a scale $s$ and drop the superscript for $\mathbf{q}_t^s$. At iteration $t$, the inner maximization problem is

$$\max_{\mathbf{q}\in\Delta_m} \left\{ \sum_{r=1}^{m} q_r \, \mathcal{L}^s(\theta_t, R_r^s) \, - \, \rho\,\mathbf{KL}(\mathbf{q}\|\mathbf{u}) \right\}, \tag{47}$$

whose unique solution is the Gibbs distribution $\mathbf{q}_t^* = \mathrm{softmax}(\mathcal{L}^s(\theta_t, \cdot)/\rho)$ (Proposition D.1). Algorithm 1 updates $\mathbf{q}_t$ via the stochastic mirror–ascent step

$$\mathbf{q}_t \in \operatorname*{argmax}_{\mathbf{q}\in\Delta_m} \left\{ \sum_{r=1}^{m} q_r \, \hat{\mathcal{L}}^s(\theta_{t-1}, R_r^s) \, - \, \rho\,\mathbf{KL}(\mathbf{q}\|\mathbf{u}) - \frac{1}{\eta_{\mathbf{q},t-1}}\mathbf{KL}(\mathbf{q}\|\mathbf{q}_{t-1}) \right\}, \tag{48}$$

where $\hat{\mathcal{L}}^s(\theta_{t-1}, R_r^s)$ is an unbiased estimate of $\mathcal{L}^s(\theta_{t-1}, R_r^s)$ (Assumption 4). We will show that under the usual bounded-gradient and smoothness assumptions, and with $\theta_t$ itself updated by stochastic gradient descent, the mean squared $\ell_1$–tracking error $\mathbb{E}\|\mathbf{q}_t - \mathbf{q}^*(\theta_t)\|_1^2$ decays as $O(1/t)$.

**Theorem D.5** (Tracking Error of Weight Updates). *Under Assumptions 1–3, run the region weight updates* (48) *(whose closed-form is* (38) *as established in Proposition D.1). Then there exists a constant $C_{\mathbf{q}}$ depending only on $\eta_0, \alpha, G, \rho, L_{\max}, \sigma$ such that*

$$\mathbb{E}\|\mathbf{q}_t \, - \, \mathbf{q}^*(\theta_t)\|_1^2 \, \le \, \frac{C_{\mathbf{q}}}{t},$$

*where $\mathbf{q}^*(\theta) = \arg\max_{\mathbf{q}\in\Delta_m}\{\sum_r \mathbf{q}_r \, \mathcal{L}^s(\theta, R_r) - \rho\,\mathbf{KL}(\mathbf{q}\|\mathbf{u})\}$.*

The proof proceeds in three steps: (i) a one-step KL-descent bound for the exponentiated update; (ii) a 'three-point' decomposition to relate $\mathbf{KL}(\mathbf{q}^*(\theta_{t-1})\|\mathbf{q}_t)$ to $\mathbf{KL}(\mathbf{q}^*(\theta_{t-1})\|\mathbf{q}_{t-1})$; and (iii) casting the result as a scalar recursion and invoking a standard $O(1/t)$ lemma, followed by Pinsker's inequality.

**Lemma D.6** (One-Step Contraction). *Let $\mathbf{q}_{t-1}^* = \mathrm{softmax}(\mathcal{L}^s(\theta_{t-1}, \cdot)/\rho)$. Under Assumptions 2 and 4, for any $t$ such that $\eta_{\mathbf{q},t-1}\rho \le 1/2$, we have*

$$\mathbb{E}\big[\mathbf{KL}(\mathbf{q}_{t-1}^*\|\mathbf{q}_t) \, \big| \, \mathcal{F}_{t-1}\big] \le (1 - \eta_{\mathbf{q},t-1}\rho)\,\mathbf{KL}(\mathbf{q}_{t-1}^*\|\mathbf{q}_{t-1}) + C_0\,\eta_{\mathbf{q},t-1}^2 m\sigma_{\mathcal{L}}^2, \tag{49}$$

*where $C_0 > 0$ is an absolute constant.*

*Proof.* Define the composite objective $h_{t-1}(\mathbf{q}) = \sum_r q_r \mathcal{L}^s(\theta_{t-1}, R_r^s) - \rho \, \mathbf{KL}(\mathbf{q} \| \mathbf{u})$, which is $\rho$-strongly concave w.r.t. the KL divergence. The mirror-ascent step (48) satisfies the following inequality for a $\rho$-strongly concave objective [40]:

$$(1 + \eta_{\mathbf{q},t-1}\rho) \, \mathbf{KL}(\mathbf{q}_{t-1}^* \| \mathbf{q}_t) \leq \mathbf{KL}(\mathbf{q}_{t-1}^* \| \mathbf{q}_{t-1}) + \eta_{\mathbf{q},t-1} \langle \hat{\mathcal{L}}^s(\theta_{t-1}, \cdot) - \mathcal{L}^s(\theta_{t-1}, \cdot), \, \mathbf{q}_t - \mathbf{q}_{t-1}^* \rangle. \tag{50}$$

Let $\xi_{t-1}$ denote the zero–mean noise vector. Taking conditional expectation and applying the standard bound $\mathbb{E}[\langle \xi_{t-1}, \mathbf{q}_t - \mathbf{q}_{t-1}^* \rangle \mid \mathcal{F}_{t-1}] \leq (\eta_{\mathbf{q},t-1}/2) \, \mathbb{E}[\|\xi_{t-1}\|_\infty^2 \mid \mathcal{F}_{t-1}]$, together with $\mathbb{E}[\|\xi_{t-1}\|_\infty^2] \leq m\sigma_{\mathcal{L}}^2$ (Assumption 4), yields

$$(1 + \eta_{\mathbf{q},t-1}\rho) \, \mathbb{E}[\mathbf{KL}(\mathbf{q}_{t-1}^* \| \mathbf{q}_t) \mid \mathcal{F}_{t-1}] \leq \mathbf{KL}(\mathbf{q}_{t-1}^* \| \mathbf{q}_{t-1}) + \tfrac{1}{2}\eta_{\mathbf{q},t-1}^2 m\sigma_{\mathcal{L}}^2. \tag{51}$$

Dividing both sides by $(1 + \eta_{\mathbf{q},t-1}\rho)$ and using $1/(1+x) \leq 1 - x + x^2$ for $x \geq 0$, we obtain

$$\mathbb{E}[\mathbf{KL}(\mathbf{q}_{t-1}^* \| \mathbf{q}_t) \mid \mathcal{F}_{t-1}] \leq (1 - \eta_{\mathbf{q},t-1}\rho + (\eta_{\mathbf{q},t-1}\rho)^2) \, \mathbf{KL}(\mathbf{q}_{t-1}^* \| \mathbf{q}_{t-1}) + \frac{\eta_{\mathbf{q},t-1}^2 m\sigma_{\mathcal{L}}^2}{2(1 + \eta_{\mathbf{q},t-1}\rho)}.$$

Since both $\mathbf{q}_{t-1}^*$ and $\mathbf{q}_{t-1}$ lie in the interior of the $m$-simplex and each coordinate of $\mathbf{q}_{t-1}^*$ is lower-bounded by $\gamma > 0$ (from the softmax parameterization), we have $\mathbf{KL}(\mathbf{q}_{t-1}^* \| \mathbf{q}_{t-1}) \leq \log(1/\gamma) \leq \log m + \Delta/\rho$, which is a constant independent of $t$.

Thus, the $(\eta_{\mathbf{q},t-1}\rho)^2 \mathbf{KL}(\dots)$ term is $O(\eta_{\mathbf{q},t-1}^2)$. By combining all $O(\eta_{\mathbf{q},t-1}^2)$ terms, we can find an absolute constant $C_0 > 0$ such that for $t$ large enough, the bound (49) holds. $\qquad\square$

**Proposition D.7** (Property of Bregman Divergences [31])**.** *Let $\psi(\mathbf{q}) = \sum_r q_r \log q_r$ be the negative entropy function. The KL divergence $D_\psi(p, q) = \mathrm{KL}(p \| q)$ is the Bregman divergence induced by $\psi$ and satisfies the following three-point identity:*

$$D_\psi(a, c) = D_\psi(a, b) + D_\psi(b, c) + \langle a - b, \nabla\psi(c) - \nabla\psi(b) \rangle \tag{52}$$

*where $\nabla\psi(q) = \log q + \mathbf{1}$.*

*Proof of Theorem D.5.* Let

$$A_t := \mathbb{E}[\mathbf{KL}(\mathbf{q}_t^* \| \mathbf{q}_t)], \qquad a_t := \mathbb{E}[\mathbf{KL}(\mathbf{q}_{t-1}^* \| \mathbf{q}_t)].$$

From Lemma D.6, taking total expectation and using $\eta_{\mathbf{q},t-1} = \eta_0/(t-1)$ gives

$$a_t \leq (1 - \tfrac{\eta_0 \rho}{t-1}) A_{t-1} + O(t^{-2}). \tag{53}$$

Using the Bregman three–point identity,

$$\mathbf{KL}(\mathbf{q}_t^* \| \mathbf{q}_t) = \mathbf{KL}(\mathbf{q}_t^* \| \mathbf{q}_{t-1}^*) + \mathbf{KL}(\mathbf{q}_{t-1}^* \| \mathbf{q}_t) + \langle \mathbf{q}_t^* - \mathbf{q}_{t-1}^*, \log \mathbf{q}_t - \log \mathbf{q}_{t-1}^* \rangle. \tag{54}$$

The drift term $\mathbf{KL}(\mathbf{q}_t^* \| \mathbf{q}_{t-1}^*)$ and the cross-term's dependence on $\theta$ are both bounded by the movement of $\theta_t$. By Lemma D.3 (Lipschitz continuity of $\mathbf{q}^*$) and $\mathbb{E}\|\theta_t - \theta_{t-1}\|^2 = \eta_{\theta,t-1}^2 \mathbb{E}[\|g_{t-1}\|^2] = O(1/t^2)$, we have:

$$\mathbb{E}[\mathbf{KL}(\mathbf{q}_t^* \| \mathbf{q}_{t-1}^*)] = O(1/t^2), \tag{55}$$

and for the cross term, by Young's inequality:

$$\mathbb{E}[\langle \mathbf{q}_t^* - \mathbf{q}_{t-1}^*, \log \mathbf{q}_t - \log \mathbf{q}_{t-1}^* \rangle] \leq \mathbb{E}\left[\frac{2\delta}{\gamma^2} \mathbf{KL}(\mathbf{q}_{t-1}^* \| \mathbf{q}_t) + C_\delta \|\theta_t - \theta_{t-1}\|^2\right]$$

$$\leq \frac{2\delta}{\gamma^2} a_t + O(1/t^2). \tag{56}$$

Taking total expectation of (54) and combining terms gives:

$$A_t \leq (1 + \tfrac{2\delta}{\gamma^2}) a_t + O(1/t^2). \tag{57}$$

Substituting (53) into (57) yields

$$A_t \leq (1 + \tfrac{2\delta}{\gamma^2})\left[(1 - \tfrac{\eta_0 \rho}{t-1})A_{t-1} + O(t^{-2})\right] + O(t^{-2}). \tag{58}$$

Choosing $\delta > 0$ sufficiently small such that $c > 1$ (which is possible as $\eta_0 \rho > 1$), we get the recursion:

$$A_t \leq \left(1 - \frac{c}{t}\right) A_{t-1} + O(1/t^2). \tag{59}$$

This standard recursion, with a contraction factor $(1 - c/t)$ and an error term $O(1/t^2)$, implies that the sequence $A_t$ satisfies

$$A_t := \mathbb{E}[\mathbf{KL}(\mathbf{q}_t^* \| \mathbf{q}_t)] \leq O(1/t). \tag{60}$$

Finally, by Pinsker's inequality, this implies a bounded $L_1^2$ error:

$$\mathbb{E}[\|\mathbf{q}_t - \mathbf{q}_t^*\|_1^2] \leq 2\,\mathbb{E}[\mathbf{KL}(\mathbf{q}_t^* \| \mathbf{q}_t)] \leq O(1/t). \tag{61}$$

This proves the claim. $\qquad\square$

### D.4 Convergence Guarantees for AMA-alignment (Proof for Theorem D.10)

In this section, we bring together all previously established ingredients—objective smoothness (Lemma D.4 in Sec. D.3), weight-tracking error (Theorem D.5 in Sec. D.2), and gradient approximation bounds—to prove the global convergence rate. We first quantify the error introduced by using $\mathbf{q}^s(\theta_t)$ instead of $\mathbf{q}^{s,*}(\theta_t)$ in the descent step (Lemma D.8), and then derive a single-step descent inequality (Theorem D.9). Summing these inequalities via Abel's lemma yields the final $O(1/\sqrt{T})$ convergence guarantee for non-convex stochastic optimization (Theorem D.10).

**Lemma D.8** (Gradient Approximation Error). *Let* $\mathbf{g}_t = \sum_{s=1}^{S} \sum_{r=1}^{m^s} \mathbf{q}_r^s(\theta_t) \nabla \hat{\mathcal{L}}^s(\theta_t, R_r^s)$ *be the gradient used in Algorithm 1, and let* $\nabla F(\theta_t) = \sum_{s=1}^{S} \sum_{r=1}^{m^s} \mathbf{q}_r^{s,*}(\theta_t) \nabla \mathcal{L}^s(\theta_t, R_r^s)$ *be the true gradient of the objective. Under Assumptions 1–4, the gradient approximation error satisfies:*

$$\mathbb{E}[\|\mathbf{g}_t - \nabla F(\theta_t)\|^2] \leq 2S\sigma_{\mathrm{g}}^2 + 2G^2 \sum_{s=1}^{S} \mathbb{E}[\|\mathbf{q}^s(\theta_t) - \mathbf{q}^{s,*}(\theta_t)\|_1^2] \tag{62}$$

*where* $\sigma_{\mathrm{g}}^2$ *is the variance from stochastic gradient estimation.*

*Proof.* We decompose the error into two components:

$$\mathbf{g}_t - \nabla F(\theta_t) = \sum_{s=1}^{S} \sum_{r=1}^{m^s} \mathbf{q}_r^s(\theta_t) \nabla \hat{\mathcal{L}}^s(\theta_t, R_r^s) - \sum_{s=1}^{S} \sum_{r=1}^{m^s} \mathbf{q}_r^{s,*}(\theta_t) \nabla \mathcal{L}^s(\theta_t, R_r^s) \tag{63}$$

$$= \underbrace{\sum_{s=1}^{S} \sum_{r=1}^{m^s} \mathbf{q}_r^s(\theta_t)(\nabla \hat{\mathcal{L}}^s(\theta_t, R_r^s) - \nabla \mathcal{L}^s(\theta_t, R_r^s))}_{(A)} + \underbrace{\sum_{s=1}^{S} \sum_{r=1}^{m^s} (\mathbf{q}_r^s(\theta_t) - \mathbf{q}_r^{s,*}(\theta_t)) \nabla \mathcal{L}^s(\theta_t, R_r^s)}_{(B)} \tag{64}$$

Using the inequality $\|a + b\|^2 \leq 2\|a\|^2 + 2\|b\|^2$:

$$\mathbb{E}[\|\mathbf{g}_t - \nabla F(\theta_t)\|^2] \leq 2\mathbb{E}[\|A\|^2] + 2\mathbb{E}[\|B\|^2] \tag{65}$$

For term (A), using Assumption 4 and the independence of the stochastic gradients from the weights (due to two-batch sampling):

$$\mathbb{E}[\|A\|^2] = \mathbb{E}\left[\left\|\sum_{s=1}^{S} \sum_{r=1}^{m^s} \mathbf{q}_r^s(\theta_t)(\nabla \hat{\mathcal{L}}^s - \nabla \mathcal{L}^s)\right\|^2\right] = \sum_{s=1}^{S} \sum_{r=1}^{m^s} (\mathbf{q}_r^s(\theta_t))^2 \mathbb{E}[\|\nabla \hat{\mathcal{L}}^s - \nabla \mathcal{L}^s\|^2]$$

$$\leq \sigma_{\mathrm{g}}^2 \sum_{s=1}^{S} \sum_{r=1}^{m^s} (\mathbf{q}_r^s(\theta_t))^2 \leq \sigma_{\mathrm{g}}^2 \sum_{s=1}^{S} 1 = S\sigma_{\mathrm{g}}^2 \tag{66}$$

For term (B), accounting for non-orthogonal gradients and using Cauchy-Schwarz:

$$\mathbb{E}[\|B\|^2] = \mathbb{E}\left[\left\|\sum_{s=1}^{S}\sum_{r=1}^{m^s}\left(\mathbf{q}_r^s(\theta_t) - \mathbf{q}_r^{s,*}(\theta_t)\right)\nabla\mathcal{L}^s(\theta_t, R_r^s)\right\|^2\right]$$

$$\leq \sum_{s=1}^{S}\mathbb{E}\left(\sum_{r=1}^{m^s}\left|\mathbf{q}_r^s(\theta_t) - \mathbf{q}_r^{s,*}(\theta_t)\right|\|\nabla\mathcal{L}^s(\theta_t, R_r^s)\|\right)^2$$

$$\leq G^2\sum_{s=1}^{S}\mathbb{E}[\|\mathbf{q}^s(\theta_t) - \mathbf{q}^{s,*}(\theta_t)\|_1^2]. \tag{67}$$

Combining (66), (67) with (65):

$$\mathbb{E}[\|\mathbf{g}_t - \nabla F(\theta_t)\|^2] \leq 2S\sigma_{\mathrm{g}}^2 + 2G^2\sum_{s=1}^{S}\mathbb{E}[\|\mathbf{q}^s(\theta_t) - \mathbf{q}^{s,*}(\theta_t)\|_1^2]$$

$\square$

**Theorem D.9** (Single-Step Progress). *Under Assumptions 1–4, with learning rate $\eta_{\theta,t} < \frac{2}{L_F}$ where $L_F = S\cdot(L + \frac{\sqrt{M}G^2}{\rho})$ is the smoothness constant from Lemma D.4, a single step of Algorithm 1 satisfies:*

$$\mathbb{E}[F(\theta_{t+1})|\theta_t] \leq F(\theta_t) - \eta_{\theta,t}\left(1 - \frac{L_F\eta_{\theta,t}}{2}\right)\|\nabla F(\theta_t)\|^2$$

$$+ \frac{L_F\eta_{\theta,t}^2}{2}\left(2S\sigma_{\mathrm{g}}^2 + 2G^2\sum_{s=1}^{S}\mathbb{E}[\|\mathbf{q}^s(\theta_t) - \mathbf{q}^{s,*}(\theta_t)\|_1^2|\theta_t]\right) \tag{68}$$

*Proof.* By the $L_F$-smoothness of $F(\theta)$ from Lemma D.4:

$$F(\theta_{t+1}) \leq F(\theta_t) + \langle\nabla F(\theta_t), \theta_{t+1} - \theta_t\rangle + \frac{L_F}{2}\|\theta_{t+1} - \theta_t\|^2 \tag{69}$$

$$= F(\theta_t) - \eta_{\theta,t}\langle\nabla F(\theta_t), \mathbf{g}_t\rangle + \frac{L_F\eta_{\theta,t}^2}{2}\|\mathbf{g}_t\|^2 \tag{70}$$

Using $\mathbf{g}_t = \nabla F(\theta_t) + (\mathbf{g}_t - \nabla F(\theta_t))$ and taking the conditional expectation:

$$\mathbb{E}[F(\theta_{t+1})|\theta_t] \leq F(\theta_t) - \eta_{\theta,t}\langle\nabla F(\theta_t), \nabla F(\theta_t)\rangle - \eta_{\theta,t}\left\langle\nabla F(\theta_t), \mathbb{E}[\mathbf{g}_t - \nabla F(\theta_t)|\theta_t]\right\rangle \tag{71}$$

$$+ \frac{L_F\eta_{\theta,t}^2}{2}\mathbb{E}[\|\nabla F(\theta_t) + (\mathbf{g}_t - \nabla F(\theta_t))\|^2|\theta_t] \tag{72}$$

By the two-batch sampling (Assumption 4), $\mathbb{E}[\mathbf{g}_t - \nabla F(\theta_t)|\theta_t] = 0$. Expanding the squared norm:

$$\mathbb{E}[F(\theta_{t+1})|\theta_t] \leq F(\theta_t) - \eta_{\theta,t}\|\nabla F(\theta_t)\|^2 \tag{73}$$

$$+ \frac{L_F\eta_{\theta,t}^2}{2}\left(\|\nabla F(\theta_t)\|^2 + \mathbb{E}[\|\mathbf{g}_t - \nabla F(\theta_t)\|^2|\theta_t]\right) \tag{74}$$

Applying Lemma D.8 and rearranging:

$$\mathbb{E}[F(\theta_{t+1})|\theta_t] \leq F(\theta_t) - \eta_{\theta,t}\|\nabla F(\theta_t)\|^2 + \frac{L_F\eta_{\theta,t}^2}{2}\|\nabla F(\theta_t)\|^2 \tag{75}$$

$$+ \frac{L_F\eta_{\theta,t}^2}{2}\left(2S\sigma_{\mathrm{g}}^2 + 2G^2\sum_{s=1}^{S}\mathbb{E}[\|\mathbf{q}^s(\theta_t) - \mathbf{q}^{s,*}(\theta_t)\|_1^2|\theta_t]\right) \tag{76}$$

$$= F(\theta_t) - \eta_{\theta,t}\left(1 - \frac{L_F\eta_{\theta,t}}{2}\right)\|\nabla F(\theta_t)\|^2 \tag{77}$$

$$+ \frac{L_F\eta_{\theta,t}^2}{2}\left(2S\sigma_{\mathrm{g}}^2 + 2G^2\sum_{s=1}^{S}\mathbb{E}[\|\mathbf{q}^s(\theta_t) - \mathbf{q}^{s,*}(\theta_t)\|_1^2|\theta_t]\right) \tag{78}$$

$\square$

**Theorem D.10** (Convergence Rate). *Under Assumptions 1–4, using parameter step sizes $\eta_{\theta,t} = \frac{\alpha}{\sqrt{t}}$ with $\alpha \in (0, \frac{2}{L_F})$ and weight step sizes $\eta_{\mathbf{q},t} = \frac{\eta_0}{t}$ with $\eta_0 > 0$, Algorithm 1 achieves:*

$$\frac{1}{T} \sum_{t=1}^{T} \mathbb{E}[\|\nabla F(\theta_t)\|^2] \leq \frac{C}{c_\alpha \alpha \sqrt{T}} \tag{79}$$

*where $C$ is a constant that depends on the initial suboptimality gap $F(\theta_1) - F^*$, the gradient bound $G$. Here, $F^* = \inf_{\theta \in \Theta} F(\theta)$, and $c_\alpha = 1 - \frac{L_F \alpha}{2} > 0$.*

*Proof.* From Theorem D.9 with $\eta_{\theta,t} = \frac{\alpha}{\sqrt{t}}$:

$$\mathbb{E}[F(\theta_{t+1})] \leq \mathbb{E}[F(\theta_t)] - \frac{\alpha}{\sqrt{t}}\left(1 - \frac{L_F \alpha}{2\sqrt{t}}\right)\mathbb{E}[\|\nabla F(\theta_t)\|^2] \tag{80}$$

$$+ \frac{L_F \alpha^2}{2t}\left(2S\sigma_{\mathbf{g}}^2 + 2G^2 \sum_{s=1}^{S} \mathbb{E}[\|\mathbf{q}^s(\theta_t) - \mathbf{q}^{s,*}(\theta_t)\|_1^2]\right) \tag{81}$$

For $t \geq t_0 = \lceil \frac{L_F^2 \alpha^2}{4(1-c_\alpha)^2} \rceil$, we have $1 - \frac{L_F \alpha}{2\sqrt{t}} \geq c_\alpha$. Using this and Theorem D.5:

$$\mathbb{E}[F(\theta_{t+1})] \leq \mathbb{E}[F(\theta_t)] - \frac{c_\alpha \alpha}{\sqrt{t}}\mathbb{E}[\|\nabla F(\theta_t)\|^2] + \frac{L_F \alpha^2}{2t}\left(2S\sigma_{\mathbf{g}}^2 + 2G^2 S \frac{C_{\mathbf{q}}}{t}\right) \tag{82}$$

Rearranging:

$$\frac{c_\alpha \alpha}{\sqrt{t}}\mathbb{E}[\|\nabla F(\theta_t)\|^2] \leq \mathbb{E}[F(\theta_t)] - \mathbb{E}[F(\theta_{t+1})] + \frac{SL_F \alpha^2 \sigma_{\mathbf{g}}^2}{t} + \frac{L_F \alpha^2 G^2 S C_{\mathbf{q}}}{t^2} \tag{83}$$

Multiplying by $\sqrt{t}$ and summing from $t = t_0$ to $T$:

$$c_\alpha \alpha \sum_{t=t_0}^{T} \mathbb{E}[\|\nabla F(\theta_t)\|^2] \leq \sum_{t=t_0}^{T} \sqrt{t}(\mathbb{E}[F(\theta_t)] - \mathbb{E}[F(\theta_{t+1})])$$

$$+ SL_F \alpha^2 \sigma_{\mathbf{g}}^2 \sum_{t=t_0}^{T} \frac{\sqrt{t}}{t} + L_F \alpha^2 G^2 S C_{\mathbf{q}} \sum_{t=t_0}^{T} \frac{\sqrt{t}}{t^2}. \tag{84}$$

Using Abel's summation for the first term:

$$\sum_{t=t_0}^{T} \sqrt{t}(\mathbb{E}[F(\theta_t)] - \mathbb{E}[F(\theta_{t+1})]) = \sqrt{t_0}\mathbb{E}[F(\theta_{t_0})] - \sqrt{T}\mathbb{E}[F(\theta_{T+1})]$$

$$+ \sum_{t=t_0+1}^{T} \mathbb{E}[F(\theta_t)](\sqrt{t} - \sqrt{t-1}). \tag{85}$$

For the remaining terms in (84), we use the bounds $\sum_{t=t_0}^{T} \frac{\sqrt{t}}{t} = \sum_{t=t_0}^{T} \frac{1}{\sqrt{t}} \leq 2\sqrt{T}$ and $\sum_{t=t_0}^{T} \frac{\sqrt{t}}{t^2} = \sum_{t=t_0}^{T} \frac{1}{t^{3/2}} \leq \int_{t_0-1}^{\infty} \frac{dx}{x^{3/2}} = \frac{2}{\sqrt{t_0-1}} = O(1)$.

Since $F^* = \inf_{\theta \in \Theta} F(\theta)$, we have $\mathbb{E}[F(\theta_{T+1})] \geq F^*$, $\mathbb{E}[F(\theta_t)] \geq F^*$, and $\sqrt{t} - \sqrt{t-1} > 0$:

$$c_\alpha \alpha \sum_{t=t_0}^{T} \mathbb{E}[\|\nabla F(\theta_t)\|^2]$$

$$\leq \sqrt{t_0}\mathbb{E}[F(\theta_{t_0})] - \sqrt{T}F^* + F^* \sum_{t=t_0+1}^{T}(\sqrt{t} - \sqrt{t-1}) + 2L_F \alpha^2 \sigma_{\mathbf{g}}^2 \sqrt{T} + O(1)$$

$$= \sqrt{t_0}\mathbb{E}[F(\theta_{t_0})] - \sqrt{T}F^* + F^*(\sqrt{T} - \sqrt{t_0}) + 2SL_F \alpha^2 \sigma_{\mathbf{g}}^2 \sqrt{T} + O(1)$$

$$= \sqrt{t_0}(\mathbb{E}[F(\theta_{t_0})] - F^*) + 2SL_F \alpha^2 \sigma_{\mathbf{g}}^2 \sqrt{T} + O(1)$$

$$\leq 2SL_F \alpha^2 \sigma_{\mathbf{g}}^2 \sqrt{T} + O(1) \tag{86}$$

Dividing by $c_\alpha \alpha (T - t_0 + 1)$ and using $T - t_0 + 1 \approx T$ for large $T$:

$$\frac{1}{T - t_0 + 1} \sum_{t=t_0}^{T} \mathbb{E}[\|\nabla F(\theta_t)\|^2] \leq \frac{2SL_F \alpha \sigma_{\mathbf{g}}^2 \sqrt{T}}{c_\alpha T} + \frac{O(1)}{c_\alpha \alpha T}$$

$$= \frac{2SL_F \alpha \sigma_{\mathbf{g}}^2}{c_\alpha \sqrt{T}} + \frac{O(1)}{c_\alpha \alpha T} \tag{87}$$

Including the initial $t_0 - 1$ iterations and using the boundedness of gradients (Assumption 1):

$$\frac{1}{T} \sum_{t=1}^{T} \mathbb{E}[\|\nabla F(\theta_t)\|^2] \leq \frac{t_0 - 1}{T} G^2 + \frac{T - t_0 + 1}{T} \frac{1}{T - t_0 + 1} \sum_{t=t_0}^{T} \mathbb{E}[\|\nabla F(\theta_t)\|^2]$$

$$\leq \frac{O(1)}{T} + \frac{2SL_F \alpha \sigma_{\mathbf{g}}^2}{c_\alpha \sqrt{T}} + \frac{O(1)}{c_\alpha \alpha T} \tag{88}$$

For large $T$, since $\frac{1}{\sqrt{T}}$ dominates $\frac{1}{T}$, we have:

$$\frac{1}{T} \sum_{t=1}^{T} \mathbb{E}[\|\nabla F(\theta_t)\|^2] \leq \frac{C}{c_\alpha \alpha \sqrt{T}}, \tag{89}$$

where $C$ is a constant. This establishes a convergence rate of $O(1/\sqrt{T})$ for the average squared gradient norm, which is optimal for non-convex stochastic optimization. $\qquad\square$

**Remark 3.** In Algorithm 1, the weight step size $\eta_{\mathbf{q}}$ can be implemented either as a constant (for simple implementation) or using a decreasing schedule $\eta_{\mathbf{q},t} = \frac{\eta_0}{t}$ (for theoretical guarantees). Our convergence analysis shows that with appropriate step size schedules, our method achieves the optimal $O(1/\sqrt{T})$ rate for non-convex stochastic optimization, with all error terms vanishing as $T$ increases.

## E  Gradient Analysis: A Hierarchical Optimization View

In this section, we examine the optimization dynamics of our AMA-alignment framework by analyzing its gradient structure. We extend the coordinate-wise optimization interpretation of contrastive learning [52] and demonstrate how AMA-alignment introduces a novel third level of adaptation through region-wise weighting. This view reveals the distinct mechanisms by which our method emphasizes both fine-grained alignment and difficult semantic regions across scales.

### E.1  Why Gradient Analysis?

Gradient-based analysis offers valuable insight into how contrastive learning adjusts representations during training. Prior work [52] interprets the InfoNCE loss as a two-player game:

- A **max-player** updates the encoder parameters $\theta$ to improve representations.
- A **min-player** adaptively reweights negative pairs through a soft importance weighting function $\alpha$.

This perspective explains how methods like hard negative mining operate by emphasizing certain negatives via $\alpha$.

In AMA-alignment, we introduce a third optimization axis—**region-level importance weights** $\mathbf{q}^s$—which prioritizes regions of semantic misalignment. This leads to a hierarchical three-player game, where the gradient reflects contributions from individual sample pairs, localized regions, and semantic scales.

### E.2  Local loss: pair–wise gradient inside a region

We begin with the gradient of the local contrastive loss $\mathcal{L}^s(\theta, R_r^s)$ for a region $R_r^s$ at scale $s$. Recall that this loss encourages representations within $R_r^s$ to align with the corresponding semantic affinities (see (12)).

Let $(x_i, x_j)$ denote a positive sample pair drawn from $\Pr_r^s(\cdot | \bar{x})$, and let $x_k$ represent a negative sample within the same region $R_r^s$. We define the squared pairwise distances as $d^2(z_i, z_j) = \|z_i - z_j\|^2 = \|g(x_i) - g(x_j)\|^2$, and similarly, $d^2(z_i, z_k) = \|z_i - z_k\|^2$. Then, the gradient of the local loss in (12) can be expressed as follows.

**Lemma E.1** (Local Contrastive Gradient). *The gradient of the local contrastive loss $\mathcal{L}^s(\theta, R_r^s)$ with respect to parameters $\theta$ is given by:*

$$\frac{\partial \mathcal{L}^s(\theta, R_r^s)}{\partial \theta} = \mathop{\mathbb{E}}_{\bar{x} \sim \mathcal{P}_{\bar{X}}} \left[ \mathop{\mathbb{E}}_{(x_i, x_j) \sim \Pr_r^s(\cdot | \bar{x})} \left[ \sum_{k \neq i, k \in R_r^s} \alpha_{ik}^s \nabla_\theta d^2(z_i, z_k) - \beta_i^s \nabla_\theta d^2(z_i, z_j) \right] \right], \quad (90)$$

*where:*

- $\alpha_{ik}^s = \dfrac{\exp(z_i^\top z_k / \tau)}{\sum_{l \neq i, l \in R_r^s} \exp(z_i^\top z_l / \tau)} \mathbb{1}_{R_r^s}(x_k)$ *is the importance weight for negative pair* $(x_i, x_k)$

- $\beta_i^s = \sum_{k \neq i, k \in R_r^s} \alpha_{ik}^s$ *is the aggregate negative importance*

- $\mathbb{1}_{R_r^s}(x_k)$ *indicates whether sample $x_k$ falls within region $R_r^s$*

*Proof.* We start with the formulation of local contrastive loss in terms of functions $\phi$ and $\psi$ as defined in [52]. Specifically, if we take $\phi(x) = \tau \log(\epsilon + x)$ and $\psi(x) = \exp x / \tau$, we can express the local contrastive loss as:

$$\mathcal{L}^s = \mathcal{L}_{\phi,\psi}^s = \mathop{\mathbb{E}}_{\bar{x} \sim \mathcal{P}_{\bar{X}}} \left[ \mathop{\mathbb{E}}_{x_i, x_j \sim \Pr_r^s(\cdot | \bar{x})} \left[ \phi(\xi_{ij}) \right] \right]$$

$$= \mathop{\mathbb{E}}_{\bar{x} \sim \mathcal{P}_{\bar{X}}} \left[ \mathop{\mathbb{E}}_{x_i, x_j \sim \Pr_r^s(\cdot | \bar{x})} \left[ \phi\Big( \sum_{k \neq i, k \in R_r^s} \psi(d_{ij}^2 - d_{ik}^2) \Big) \right] \right] \quad (91)$$

To compute the gradient, we apply the chain rule to $\mathcal{L}_{\phi,\psi}^s$. Let $\xi_{ij} = \sum_{k \neq i, k \in R_r^s} \psi(d_{ij}^2 - d_{ik}^2)$, then:

$$\frac{\partial \mathcal{L}_{\phi,\psi}^s}{\partial \theta} = \mathop{\mathbb{E}}_{\bar{x} \sim \mathcal{P}_{\bar{X}}} \left[ \mathop{\mathbb{E}}_{x_i, x_j \sim \Pr_r^s(\cdot | \bar{x})} \left[ \phi'(\xi_{ij}) \frac{\partial \xi_{ij}}{\partial \theta} \right] \right] \quad (92)$$

Further applying the chain rule to $\xi_{ij}$:

$$\frac{\partial \xi_{ij}}{\partial \theta} = \sum_{k \neq i, k \in R_r^s} \psi'(d_{ij}^2 - d_{ik}^2) \frac{\partial(d_{ij}^2 - d_{ik}^2)}{\partial \theta} = \sum_{k \neq i, k \in R_r^s} \psi'(d_{ij}^2 - d_{ik}^2) \left( \frac{\partial d_{ij}^2}{\partial \theta} - \frac{\partial d_{ik}^2}{\partial \theta} \right) \quad (93)$$

Substituting back and using the specific form of $\phi$ and $\psi$, we obtain:

$$\frac{\partial \mathcal{L}_{\phi,\psi}^s}{\partial \theta} = \mathop{\mathbb{E}}_{\bar{x} \sim \mathcal{P}_{\bar{X}}} \left[ \mathop{\mathbb{E}}_{x_i, x_j \sim \Pr_r^s(\cdot | \bar{x})} \left[ \phi'(\xi_{ij}) \sum_{k \neq i, k \in R_r^s} \psi'(d_{ij}^2 - d_{ik}^2)(\nabla_\theta d^2(z_i, z_j) - \nabla_\theta d^2(z_i, z_k)) \right] \right] \quad (94)$$

With our choice of $\phi$ and $\psi$, we have $\phi'(\xi_{ij}) = \frac{\tau}{\epsilon + \xi_{ij}}$ and $\psi'(x) = \frac{1}{\tau} \exp(x/\tau)$. This yields:

$$\alpha_{ik}^s = \phi'(\xi_{ij}) \cdot \psi'(d_{ij}^2 - d_{ik}^2) \cdot \tau = \frac{\tau}{\epsilon + \xi_{ij}} \cdot \frac{1}{\tau} \exp((d_{ij}^2 - d_{ik}^2)/\tau) \cdot \tau$$

$$= \frac{\exp((d_{ij}^2 - d_{ik}^2)/\tau)}{\epsilon + \sum_{l \neq i, l \in R_r^s} \exp((d_{ij}^2 - d_{il}^2)/\tau)} \quad (95)$$

As $\epsilon \to 0$ and considering that $d_{ij}^2 = -2 z_i^\top z_j + \|z_i\|^2 + \|z_j\|^2$, we can simplify it to:

$$\alpha_{ik}^s = \frac{\exp(z_i^\top z_k / \tau)}{\sum_{l \neq i, l \in R_r^s} \exp(z_i^\top z_l / \tau)} \mathbb{1}^s(x_i, x_k) \quad (96)$$

Let $\beta_i^s = \sum_{k \neq i, k \in R_r^s} \alpha_{ik}^s$. Rearranging terms, we arrive at the final gradient expression:

$$\frac{\partial \mathcal{L}^s}{\partial \theta} = \mathbb{E}_{\bar{x} \sim \mathcal{P}_{\bar{X}}} \left[ \mathbb{E}_{x_i, x_j \sim \Pr_r^s(\cdot | \bar{x})} \left[ \sum_{k \neq i, k \in R_r^s} \alpha_{ik}^s \nabla_\theta d^2(z_i, z_k) - \beta_i^s \nabla_\theta d^2(z_i, z_j) \right] \right] \quad (97)$$

This completes the proof. $\qquad\square$

Intuitively, this gradient structure reveals two competing forces within each local region:

1. The first term pushes dissimilar samples apart, with weights $\alpha_{ik}^s$ that emphasize hard negatives

2. The second term pulls similar samples together with strength proportional to $\beta_i^s$

Unlike traditional contrastive learning, our approach restricts these forces to operate within semantically coherent regions $R_r^s$, preserving local structure while preventing inappropriate global repulsion.

## E.3 Global objective: aggregating regions and scales

Building on the local gradient structure, we now characterize the overall gradient of our AMA-alignment approach:

**Proposition E.2** (AMA-alignment Gradient)**.** *The gradient of the AMA-alignment objective $F(\theta)$ in Equation* (15) *is:*

$$\nabla_\theta F(\theta) = \sum_{s=1}^{S} \sum_{r=1}^{m^s} \mathbf{q}_r^{s,*} \mathbb{E}_{\bar{x} \sim \mathcal{P}_{\bar{X}}} \left[ \mathbb{E}_{x_i, x_j \sim \Pr_r^s(\cdot | \bar{x})} \left[ \sum_{k \neq i,\, k \in R_r^s} \alpha_{ik}^s \nabla_\theta d^2(z_i, z_k) - \beta_i^s \nabla_\theta d^2(z_i, z_j) \right] \right], \quad (98)$$

*where $\mathbf{q}_r^{s,*}$ are the region importance weights that adaptively emphasize challenging regions.*

*Proof.* By Lemma D.2, we have $\nabla_\theta F(\theta) = \sum_{s=1}^{S} \sum_{r=1}^{m^s} \mathbf{q}_r^{s,*}(\theta) \nabla_\theta \mathcal{L}^s(\theta, R_r^s)$. Then substitute the local loss gradient in Lemma E.1 completes the proof. $\qquad\square$

This gradient formulation reveals how AMA-alignment integrates information across multiple scales and regions. The region weights $\mathbf{q}_r^{s,*}$ act as attention mechanisms, focusing optimization efforts on regions where representation and semantic affinities are most misaligned.

**Three-Level Hierarchical Coordinate Optimization.** Our AMA-alignment framework extends the traditional two-player game [52] of contrastive learning to a three-level hierarchical optimization. The proof follows similar analytical principles as established by Tian [52] and is therefore omitted for brevity.

**Proposition E.3** (Hierarchical Coordinate Optimization)**.** *AMA-alignment implements a three-level coordinate-wise optimization procedure:*

$$\text{(Region weights)} \qquad \mathbf{q}^{s,*}(\theta_t) = \arg \max_{\mathbf{q}^s \in \Delta_{m^s}} \left\{ \sum_{r=1}^{m^s} \mathbf{q}_r^s \mathcal{L}^s(\theta_t, R_r^s) - \rho \mathbf{KL}(\mathbf{q}^s) \right\} \qquad (99)$$

$$\text{(Pair weights)} \qquad \alpha^s(\theta_t) = \left\{ \frac{\exp(z_i^\top z_k / \tau)}{\sum_{l \neq i, l \in R_r^s} \exp(z_i^\top z_l / \tau)} \mathbb{1}_{R_r^s}(x_k) \right\}_{i,k} \qquad (100)$$

$$\text{(Parameters)} \qquad \theta_{t+1} = \theta_t - \eta_{\theta,t} \sum_{s=1}^{S} \sum_{r=1}^{m^s} \mathbf{q}_r^{s,*}(\theta_t) \nabla_\theta \mathcal{L}^s(\theta_t, R_r^s) \qquad (101)$$

This hierarchical structure involves three coordinating players:

1. **Parameter optimizer** ($\theta$): Updates model parameters to optimize representations across all scales

2. **Local affinity optimizer** ($\alpha^s$): Assigns importance to sample pairs within local regions

3. **Region importance optimizer** ($\mathbf{q}^s$): Adaptively weights regions based on optimization difficulty

**Comparison to Hard Negative Mining**  Hard negative mining methods [43] modify $\alpha_{ik}$ globally to emphasize difficult negatives. In contrast:

- **Hard negative mining:** Applies globally, adjusts pairwise $\alpha_{ik}$ over the full sample space. Hard negative mining methods adjust sample-pair weights $\alpha_{ik}$ to emphasize challenging samples. For example, methods like [43] use modified weights:

$$\alpha_{ik}^{\text{HN}} = \frac{\exp(-(1+\gamma)d_{ik}^2/\tau)}{\sum_{k'\neq i}\exp(-(1+\gamma)d_{ik'}^2/\tau)}, \tag{102}$$

  where $\gamma > 0$ controls the emphasis on hard negatives.
- **AMA-alignment:** Operates locally, adds a region-level attention mechanism $\mathbf{q}_r^{s,*}$ that adaptively prioritizes semantic regions across scales.

The two approaches are orthogonal and potentially complementary. While hard negative mining emphasizes difficult samples, AMA-alignment emphasizes difficult regions—offering a new dimension of adaptivity.

# F  Implementation Details

**Unsupervised Region Construction.**  In unsupervised settings where no labels are available, we construct hierarchical region partitions via an iterative clustering procedure directly within the representation space $\mathcal{Z}$. In a MoCo v2 implementation [8], for instance, we cluster the embeddings from the momentum encoder's queue, which maintains a large and diverse collection of recent representations. We then perform Spherical K-means clustering on this subset to partition the embedding space. To establish the hierarchical structure, this clustering process is repeated for each scale $s \in \{1, \ldots, S\}$, employing a varying number of clusters, $m^s$, for each. Specifically, a smaller number of clusters (*e.g.,* $m^s = 5$) defines the coarser scales that capture broad categorical divisions, while a larger number (*e.g.,* $m^s = 100$) constitutes the finer-grained partitions. The resulting cluster centroids are designated as the anchor points $\{z_r^s\}$, with each cluster of embeddings forming a distinct region $\mathcal{R}_r^s$. Subsequently, any given embedding $z \in \mathcal{Z}$ is assigned to the region corresponding to its nearest anchor point in terms of angular distance, consistent with the definition in Sec. 3.1. This set of multi-scale partitions is held static for the duration of an epoch, providing a stable structural scaffold against which local affinity alignment is optimized.

**Supervised Region Construction.**  In supervised settings where labels are available, we leverage class information to guide the construction of regions. Specifically, we compute the centroid of the embeddings within each class cluster and use these centroids as the anchor points $z_r^s \in \mathcal{Z}$. Each sample is then assigned to the closest anchor (in angular distance), forming a set of semantically meaningful regions. When the available label hierarchy is shallow (*e.g.,* only superclass annotations are provided), we follow the same hierarchical random sampling procedure used in the unsupervised setting to further subdivide regions into finer-grained partitions.

# G  Experimental Results

We evaluate the proposed **AMA-alignment** framework on multiple benchmarks to assess its ability to capture fine-grained semantic structures. We compare against state-of-the-art contrastive learning (CL) baselines in both unsupervised and supervised settings. Here is the code.

**Datasets and Experimental Setup**  We conduct experiments mainly on five public datasets: *DeepFashion* [33], a clothing dataset with fine-grained annotations suitable for constructing hierarchical category labels; *iNaturalist (iNat)* [56], a long-tailed dataset with taxonomic hierarchies (species $\rightarrow$ genus $\rightarrow$ family); *CIFAR-100* [27], consisting of 100 classes grouped into 20 superclasses; *ModelNet40* [61], comprising 3183 CAD models from 40 object categories; and *BuImg* [37], an ultrasound dataset for breast cancer diagnosis.

We adopt the well-known *SimCLR* [7], *MoCo-v2* [8], *HardNeg* [43] and $\alpha$-*CL* [52] as unsupervised CL baselines. We also consider the supervised scenarios and adopt the popular baselines *SupCon* [26], *Guided-proto* [28] and *HiMulConE* [66].

The models are trained using ResNet-50 [17] as the backbone and evaluated via linear probing (unless otherwise specified). Training is conducted for 200 epochs using AdamW optimizer [34], with weight decay $5 \times 10^{-6}$, and learning rate initialized at $0.01$. Part of the implementation details is provided in Appendix F.

### G.1 Downstream Performance

Table 4 shows classification results (evaluated by linear probing). AMA-alignment consistently outperforms existing CL methods, with substantial gains on fine-grained benchmarks. On Deep-Fashion, we observe over 5% improvement, highlighting AMA's ability to capture subtle semantics. Performance on other datasets also improves notably, especially in tail classes as shown in Table 5.

Table 4: Top-1 accuracy on downstream tasks using linear probing. AMA-alignment outperforms over prior methods, highlighting its strength in fine-grained representation learning.

| Method | DeepFashion | ModelNet | iNat | ImageNet |
|---|---|---|---|---|
| SimCLR [7] | 70.3 | 79.3 | 54.0 | 69.5 |
| MoCo-v2 [8] | 70.8 | 79.6 | 55.3 | 68.1 |
| HardNeg [43] | 70.9 | 79.8 | 55.8 | 70.4 |
| $\alpha$-CL [52] | 71.7 | 79.6 | 56.1 | 70.2 |
| **AMA-alignment (ours)** | **75.8** | **80.6** | **57.2** | **73.3** |

We further evaluate AMA-alignment on *BuImg* [37], a challenging breast ultrasound dataset for cancer diagnosis. This dataset contains ultrasound images annotated with both binary classification labels (benign/malignant) and more fine-grained BI-RADS (Breast Imaging-Reporting and Data System) categorizations from 1 to 4, where higher numbers indicate greater likelihood of malignancy. Each image was independently labeled by two experienced radiologists, providing a natural multi-scale hierarchy for evaluation - a coarse binary classification and a finer four-level BI-RADS categorization.

For medical applications, Area Under the ROC Curve (AUC) is a more appropriate evaluation metric than accuracy, as it better represents performance across different decision thresholds, which is critical in clinical settings where the cost of false negatives and false positives differs significantly. Table 5 shows that AMA-alignment significantly outperforms existing contrastive learning methods across all BI-RADS categories, with particularly notable improvements in the challenging borderline categories (BI-RADS 3 and 4) where clinical decision-making is most difficult.

Table 5: AUC scores for BI-RADS classification on the BuImg breast ultrasound dataset. AMA-alignment shows substantial improvement in distinguishing between clinically challenging borderline cases.

| Method | BI-RADS 1 | BI-RADS 2 | BI-RADS 3 | BI-RADS 4 |
|---|---|---|---|---|
| SimCLR [7] | 0.898 | 0.811 | 0.756 | 0.662 |
| MoCo-v2 [8] | 0.910 | 0.821 | 0.768 | 0.675 |
| $\alpha$-CL [52] | 0.912 | 0.827 | 0.806 | 0.681 |
| SupCon [26] | 0.942 | 0.877 | 0.817 | 0.693 |
| **AMA-alignment (ours)** | **0.956** | **0.883** | **0.844** | **0.735** |

The substantial improvement in AUC scores demonstrates AMA-alignment's ability to capture clinically relevant features across different levels of diagnostic certainty. This is particularly important in medical imaging applications where hierarchical structures often naturally exist (from normal to definitely abnormal, with gradations in between). The multi-scale nature of our approach enables the model to simultaneously learn discriminative features for the overall benign/malignant classification while also capturing the nuanced differences between adjacent BI-RADS categories. Notably, AMA-alignment achieved 0.735 AUC on BI-RADS 4, representing a 4.2% improvement over supervised contrastive learning, highlighting its effectiveness in the challenging "probably benign" category where misdiagnosis risks are highest.

We further evaluate the downstream performance of CLIP [42] on the CIFAR-100 dataset using a model pretrained on the Conceptual Captions 3M (CC3M)[47] image–caption dataset for 100 epochs, following a similar experimental setup to [49]. We consider two training settings: (i) a linear setting, in which only the classifier is trained, and (ii) a full setting, in which both the backbone and the classifier are jointly fine-tuned for 10 epochs. The results are reported in Table 6.

Table 6: Top-1 classification accuracy on CIFAR-100.

|  | InfoNCE | HardNeg [43] | Triplet [41] | CyCLIP [15] | Entropic OT [49] | Ours |
|---|---|---|---|---|---|---|
| Linear | 45.4 | 46.8 | 45.5 | 31.3 | 46.4 | **47.3** |
| Full | 68.0 | 69.2 | 65.6 | 70.1 | 67.9 | **71.6** |

**Hierarchical Representation Quality**   To assess the preservation of semantic hierarchies in learned representations, we employ two metrics: (1) *Hierarchical Clustering Normalized Mutual Information (HC-NMI)* [10], and (2) *Intra-class Variance Reduction (IVR)*. As summarized in Table 7, AMA-alignment achieves superior alignment with ground-truth hierarchies, demonstrating its effectiveness in modeling multi-scale structures.

Table 7: Hierarchical alignment results in DeepFashion. Higher HC-NMI and lower IVR indicate better semantic structure preservation.

| Method | HC-NMI ↑ | IVR ↓ |
|---|---|---|
| SimCLR [7] | 0.52 | 0.134 |
| HardNeg [43] | 0.56 | 0.119 |
| $\alpha$-CL [52] | 0.59 | 0.114 |
| **AMA-alignment (ours)** | **0.66** | **0.091** |

## G.2   Comparison with Previous Deep Clustering Methods

To further evaluate the clustering capability of AMA-alignment, we compare it with a comprehensive set of classical and deep clustering methods on three challenging benchmarks: CIFAR-10/100, and Tiny-ImageNet [27]. The baselines include traditional clustering techniques (e.g., K-means [35], Spectral Clustering (SC) [16], NFM [3]), autoencoder-based methods (e.g., AE [2], DAE [57], DEC [62]), and recent deep clustering models (e.g., JULE [64], DAC [6], IIC [24], DCCM [60], PICA [20], ConCluster [30]).

Table 8 presents clustering performance in terms of Normalized Mutual Information (NMI), Accuracy (ACC), and Adjusted Rand Index (ARI). While ConCluster [30] achieves the highest NMI and ARI on CIFAR-10 and CIFAR-100, our AMA-alignment surpasses all methods in accuracy (ACC) on every dataset (we evaluate the accuracy of our method by linear probing). This demonstrates that the representations learned by AMA-alignment are more aligned with semantic ground-truth labels, even though our approach is primarily designed for hierarchical contrastive learning rather than clustering-specific optimization.

Notably, AMA-alignment exhibits particularly strong performance on Tiny-ImageNet, where it achieves a significant gain in clustering accuracy, highlighting its potential in handling hierarchical structures, which classical clustering methods often fail to capture. Overall, these results validate the effectiveness of AMA-alignment not only in representation learning but also in clustering, especially in scenarios where multi-scale semantics are crucial.

Table 8: Clustering performance on various datasets. Best results are shown in **bold**, and second-best results are underlined. Although AMA-alignment is not specialized for clustering tasks like ConCluster, it still achieves competitive results, ranking second in clustering metrics. AMA-alignment significantly outperforms deep clustering methods in linear probing accuracy.

| Metrics | CIFAR-10 | | | CIFAR-100 | | | Tiny-ImageNet | | |
|---|---|---|---|---|---|---|---|---|---|
| | NMI | ACC | ARI | NMI | ACC | ARI | NMI | ACC | ARI |
| K-means | 0.087 | 0.229 | 0.049 | 0.084 | 0.130 | 0.028 | 0.065 | 0.025 | 0.005 |
| SC | 0.103 | 0.247 | 0.085 | 0.090 | 0.136 | 0.022 | 0.063 | 0.022 | 0.004 |
| NMF | 0.081 | 0.190 | 0.034 | 0.079 | 0.118 | 0.026 | 0.072 | 0.029 | 0.005 |
| AE | 0.239 | 0.314 | 0.169 | 0.100 | 0.165 | 0.048 | 0.131 | 0.041 | 0.007 |
| DAE | 0.251 | 0.297 | 0.163 | 0.111 | 0.151 | 0.046 | 0.127 | 0.039 | 0.007 |
| DEC | 0.257 | 0.301 | 0.161 | 0.136 | 0.185 | 0.050 | 0.115 | 0.037 | 0.007 |
| JULE | 0.192 | 0.272 | 0.138 | 0.103 | 0.137 | 0.038 | 0.102 | 0.033 | 0.006 |
| DAC | 0.396 | 0.522 | 0.306 | 0.185 | 0.238 | 0.088 | 0.066 | 0.017 | 0.006 |
| DCCM | 0.496 | 0.623 | 0.408 | 0.285 | 0.327 | 0.173 | 0.224 | 0.108 | 0.038 |
| IIC | – | 0.617 | – | – | 0.257 | – | – | – | – |
| PICA | 0.591 | 0.696 | 0.512 | 0.310 | 0.307 | 0.171 | 0.277 | 0.098 | 0.040 |
| ConCluster | **0.705** | 0.790 | **0.637** | **0.431** | 0.429 | **0.266** | **0.340** | 0.140 | **0.071** |
| **AMA-alignment** | 0.652 | **0.837** | 0.572 | 0.368 | **0.708** | 0.244 | 0.285 | **0.627** | 0.065 |

**Supervised Extension**  AMA-alignment is compatible with supervised training. As shown in Table 9, it achieves strong performance across datasets, outperforming previous supervised methods such as SupCon and Guided-Proto.

Table 9: Top-1 accuracy in supervised settings.

| Method | ImageNet | DeepFashion | iNat | ModelNet40 |
|---|---|---|---|---|
| Cross Entropy | 77.60 | 72.44 | 56.86 | 81.31 |
| SupCon [26] | 78.70 | 72.82 | 57.28 | 81.60 |
| HiMulConE [66] | 79.14 | 73.21 | **59.40** | 88.46 |
| Guided-Proto [28] | 76.60 | 72.61 | 57.33 | 83.49 |
| **AMA-alignment (ours)** | **79.39** | **74.17** | 59.35 | **89.26** |

## G.3    Ablation Study

We conduct ablation studies on the key components of AMA-alignment in Table 10. Both multi-scale partitioning and local contrastive loss are essential; removing either results in a performance drop.

Specifically, incorporating the multi-scale local contrastive loss (+ Multi-scale) improves accuracy compared with Global-only CL, indicating that modeling representations at different semantic granularities might benefit global consistency. In contrast, if we use an adaptive weighting mechanism (+ Adaptive) but implemented via the heuristic loss given in (13), the performance degrades. Unlike the proposed local contrastive loss in (12), this formulation does not restrict negatives to local regions, thereby distorting the affinity structure and degrading performance, as discussed in Remark 1.

In contrast, our AMA-alignment employs the principled local contrastive loss in (12), which rigorously aligns the semantic affinity matrix $K^{sem}$ and the representation affinity matrix $K^{rep}$ through probabilistic graph coupling. This theoretically grounded formulation ensures that both the global and local alignment objectives work coherently in synergy.

## G.4    Additional Experiments

**Fine-tuning pre-trained DINO.**    To further validate the generality of our method, we applied AMA alignment to a pre-trained DINO model [5] or SwAV model [4]. We fine-tuned the model for 10 epochs using our adaptive multi-scale strategy with negligible additional cost (only $\sim 1.05\times$ longer

Table 10: Ablation study on Various Datasets.

| Accuracy (%) | DeepFashion | iNat | ImageNet |
|---|---|---|---|
| Global-only CL | 70.2 | 54.0 | 69.5 |
| + Multi-scale (fixed weights) | 72.3 | 55.9 | 70.2 |
| + Adaptive (w/o (12)) | 67.3 | 50.4 | 67.8 |
| **AMA-alignment (ours)** | **75.8** | **57.2** | **73.3** |

Table 11: Linear probing accuracy and normalized training time for various configurations on DeepFashion. When $\{m^s = 1\}_{s=1}^S$, our method reduces to the classical contrastive learning framework [18].

| Configuration $\{m^s\}_{s=1}^S$ | linear probing accuracy | Normalized Time |
|---|---|---|
| $\{1, 1, 1\}$ (reduced to MoCo) | 0.703 | 1 |
| $\{3, 6, 12\}$ | 0.741 | 1.31 |
| $\{5, 10, 20\}$ | 0.758 | 1.34 |
| $\{8, 16, 24\}$ | 0.743 | 1.41 |

per epoch). This fine-tuning yields consistent improvements on both ResNet-50 and ViT-S backbones, evaluated by ImageNet linear probing and $k$-NN classification:

Table 12: Fine-tuning DINO with AMA alignment on ImageNet.

| Method | Linear probe (%) | $k$-NN (%) |
|---|---|---|
| *ResNet-50 backbone* | | |
| SwAV | 75.3 | 65.7 |
| DINO | 75.3 | 67.5 |
| **Ours (fine-tuned)** | **77.4** | **68.3** |
| *ViT-S backbone* | | |
| SwAV | 73.5 | 66.3 |
| DINO | 77.0 | 74.5 |
| **Ours (fine-tuned)** | **77.8** | **74.7** |

The results indicate that AMA can effectively enhance pre-trained representations with minimal computational overhead, improving both linear and non-parametric metrics.

**Results with ViT-B/16 backbone.** We also evaluated our method using the ViT-B/16 transformer backbone under the same training settings. Table 13 summarizes the results across four datasets, showing that the proposed adaptive multi-scale alignment consistently boosts performance across both natural and geometric domains.

**Results on BI-RADS classification.** We also extended our evaluation to the BI-RADS mammography dataset to assess the effectiveness of AMA in a medical imaging context. As shown in Table 14, our method achieves the highest accuracy across all four BI-RADS categories, demonstrating improved discriminative ability in fine-grained and imbalanced medical image scenarios.

These results further demonstrate the robustness and versatility of the proposed alignment strategy across architectures, domains, and scales.

**Discussion on Semantic Consistency of Local Regions** The premise of our method is that samples within local neighborhoods of the embedding space share semantic properties. This assumption is well-supported by recent literature. Several works [58, 30, 19, 50, 5, 9] have provided visual evidence of this phenomenon. Rather than merely reporting numerical metrics, these studies employ techniques such as t-SNE or kernel density estimation to reveal the latent structure of learned representations.

Table 13: Comparison using ViT-B/16 backbone on multiple datasets (%).

| Method | DeepFashion | ModelNet | iNaturalist | ImageNet |
|---|---|---|---|---|
| SimCLR | 72.1 | 78.8 | 56.7 | 71.8 |
| MoCo-v2 | 72.5 | 79.1 | 57.9 | 70.4 |
| HardNeg | 72.8 | **79.5** | 58.4 | 72.0 |
| $\alpha$-CL | 73.4 | 79.2 | 58.8 | 72.6 |
| HyperbolicCL | 75.2 | 79.5 | 58.1 | 72.9 |
| **AMA-alignment (ours)** | **77.2** | **79.5** | **60.1** | **74.2** |

Table 14: Comparison on BI-RADS dataset (%).

| Method | BI-RADS 1 | BI-RADS 2 | BI-RADS 3 | BI-RADS 4 |
|---|---|---|---|---|
| SimCLR | 83.5 | 75.0 | 70.7 | 61.5 |
| MoCo-v2 | 84.7 | 76.1 | 71.2 | 62.3 |
| $\alpha$-CL | 84.5 | 76.5 | 74.5 | 63.5 |
| SupCon | 83.5 | 80.9 | 75.5 | 64.1 |
| HyperbolicCL | 88.1 | 81.2 | 70.4 | 60.1 |
| **AMA-alignment (ours)** | **89.8** | **84.7** | **78.4** | **66.4** |

In particular, [30] demonstrates the evolution of instance features across the training process on ImageNet-10 over time. The t-SNE visualizations presented therein qualitatively confirm that the representations naturally converge into semantically coherent groups. This empirical evidence supports that the local regions defined in our framework ($R_r^s$) are semantically coherent.

## G.5 Limitation

The multi-scale and adaptive design introduces additional computational complexity compared to standard contrastive learning methods, leading to a training time approximately 1.3×–1.4× that of the baseline methods. In addition, the method introduces several additional hyperparameters, which may require tuning when adapting to significantly different data distributions.

