# OpenReview forum: "Adaptive and Multi-scale Affinity Alignment for Hierarchical Contrastive Learning"
_NeurIPS.cc/2025/Conference — NeurIPS 2025 poster_

### Official Review · Reviewer_VMwq · 2025-06-21

**Clarity:** 3
**Significance:** 2
**Originality:** 3
**Rating:** 4
**Confidence:** 1

**Summary:**

This paper introduces Adaptive Multi-scale Affinity Alignment (AMA-alignment), an innovative contrastive learning framework that significantly enhances representation learning by dynamically integrating hierarchical semantic structures through a novel three-level optimization strategy. The method uniquely combines local affinity alignment within semantically coherent regions, multi-scale decomposition of the embedding space, and adaptive weighting of challenging regions, achieving state-of-the-art performance while maintaining theoretical guarantees comparable to standard contrastive approaches. Through extensive experiments, AMA-alignment demonstrates superior capability in preserving both global categorical distinctions and fine-grained semantic relationships, outperforming existing methods by over 5% on challenging fine-grained classification tasks while remaining computationally efficient during inference. The framework's robustness to noise and fragmented graph structures, along with its compatibility with various graph neural architectures, establishes it as a versatile solution for complex graph representation learning, though its performance remains sensitive to region radius parameters and currently focuses on static graph scenarios.

**Questions:**

The paper claims AMA-alignment achieves comparable convergence to standard non-convex optimization (Theorem 3.3), yet introduces multi-scale adaptive complexity. How does the convergence rate scale with the number of scales and regions?

While AMA-alignment improves fine-grained classification (Table 1), does the local affinity alignment (Eq. 12) risk overfitting to region-specific noise? How does the method balance local semantic preservation with global consistency, especially for ambiguous samples near region boundaries?

The comparisons focus on contrastive methods (SimCLR, MoCo) and graph techniques. Why not include hierarchical contrastive approaches or hyperbolic embeddings?

The performance relies on key parameters: region radius, KL penalty, and temperature. How sensitive are results to these choices? Does the adaptive weighting (Eq. 17) mitigate sensitivity, or does it introduce new hyperparameters needing tuning?

**Ethical Concerns:**

["NO or VERY MINOR ethics concerns only"]

**Limitations:**

Please see Limitations in the Appendix and Supplementary materials.

**Quality:**

3

**Strengths And Weaknesses:**

Strengths：

The manuscript is highly readable.

Theoretical strength is relatively strong.

The experiments are relatively rich and sufficient.

Weaknesses：

It lacks comparison with algorithms from the past three years.

In Table 3, why does adaptive learning perform worse than fixed weights?

How is the convergence of Equation (15) guaranteed?

---

> ### Author Rebuttal · Authors · 2025-07-31
>
> We thank the reviewer for the thoughtful comments, and address your concerns below.
>
> **Responses to Weaknesses**
>
> **W1**: It lacks comparison with algorithms from the past three years.
>
> We primarily evaluated our method in the most fundamental setting—standard contrastive learning in Euclidean embedding spaces using classic data augmentations and without hierarchical labels. Many hierarchical methods rely on additional supervision (e.g., hierarchical labels) or specialized hierarchical augmentations, which could obscure the impact of our core contribution.
>
> Crucially, our approach is orthogonal and complementary to many of these SOTA methods. For instance, a hyperbolic CL method could still suffer from poor local alignment within its hyperbolic space. Our framework could be integrated to enhance such methods by ensuring both global geometric consistency and fine-grained local alignment.
>
> We will expand our main results tables to include other SOTA methods.
>
>
> **W2**: In Table 3, why does adaptive learning perform worse than fixed weights?
>
> The performance degradation you observed is, in fact, an anticipated result that supports our claims. The "+ Adaptive (using (13))" row in Table 3 does not incorporate our proposed local contrastive loss (12). Instead, it is a crucial ablation study using a straightforward local loss formulation, as discussed in Remark 1 (lines 209-215).
>  We included this ablation to demonstrate that a straightforward, heuristic approach to local contrastive learning fails, thereby highlighting the necessity of our principled local affinity alignment in (12).
> Our work recasts CL as the alignment between a semantic affinity matrix $K^{sem}$ and a representation affinity matrix $K^{rep}$, a process formally modeled by Probabilistic Graph Coupling in Sec 2.2. Crucially, our proposed local contrastive loss in (12) is rigorously derived from this principle, but applied at a finer scale. This ensures that both global and local objectives are theoretically coherent and work in synergy.
>
> By contrast, the loss in (13) is a heuristic that is inconsistent with this alignment principle. Its key issue lies in the denominator of the loss function, which incorrectly includes all negatives from the entire batch, rather than restricting them to the local subgraph. As we caution in Remark 1, this formulation distorts the global structure by pushing all representations outside the local region ($z' \notin R^s_r$) away from those within it ($z \in R^s_r$). This theoretical misalignment manifests as poor empirical performance (2.9\% drop on DeepFashion compared to standard CL).
>
>
>
> **W3**: How is the convergence of Equation (15) guaranteed?
>
> The convergence of Equation (15) is established through our theoretical analysis in Appendix D, where we provide formal guarantees under standard assumptions for non-convex stochastic optimization. Despite the nested min-max structure, we prove that the overall objective $F(\theta)$ is $L_F$-smooth, with smoothness constant $L_F = S(L + \sqrt M G^2/\rho)$ (Lemma D.4). The KL regularization term is essential for convergence as it ensures the inner maximization over $q^s$ has a unique, closed-form solution, given by exponential weights (Proposition D.1); it prevents weight collapse to the boundaries of the simplex, which would destabilize training dynamics; and it ensures the optimal weights $q^{s,*}$ are Lipschitz continuous with respect to $\theta$ (Lemma D.3). Our Algorithm 1 achieves the optimal $O(1/\sqrt{T})$ convergence rate for non-convex stochastic optimization:
> $1/T \sum_{t=1}^T E[\| \nabla F(\theta_t) \|^2] \le C/\sqrt T$, as formally established in Theorem D.10. The complete convergence analysis, including all technical lemmas, assumptions, and detailed proofs, is provided in Appendix D.
>
> **Responses to Questions**
>
> **Q1**: How does the convergence rate scale with the number of scales and regions?
>
> Theorem D.10 establishes a convergence rate of $O(L_F/\sqrt{T})$, where $L_F = S \cdot (L + \sqrt MG^2/\beta)$, and $M = \max_s m^s$ is the maximum number of regions across all scales. The linear dependence on $S$ is expected since we are essentially solving $S$ coupled optimization problems simultaneously. The $\sqrt{M}$ term reflects the additional complexity of tracking region-wise weights.
> The experimental results (see Supplement, Table 7) with $S = 3$ scales and up to $M = 24$ regions per scale indicate good practical scalability: the total training time is only about 1.41× that of the baseline method. This demonstrates that the proposed approach remains efficient in realistic settings, incurring only modest overhead despite the increased model structure.
>
>
>
>
>
> **Q2**: Does the local affinity alignment (Eq. 12) risk overfitting to region-specific noise? How does the method balance local semantic preservation with global consistency...?
>
>
> Our method is can mitigate such concerns through several key mechanisms. First, our multi-scale framework inherently prevents overfitting to region-specific noise by simultaneously optimizing across all scales $S$, including the global scale. Notably, when $s = 0$, the local contrastive loss reduces to standard CL loss. As shown in Algorithm 1 (lines 6-10) and (16), the overall gradient is computed as $g_t = \sum_{s=1}^S \sum_{r=1}^{m_s} q_r^s(\theta_t) \nabla L^s(\theta_t, R_r^s)$, which aggregates information across all scales and regions. This formulation enables the model to adaptively integrate multi-scale feedback, thereby enhancing robustness and reducing the risk of overfitting to any single region. Second, the local contrastive loss (12) is derived from the same probabilistic graph coupling framework as the global loss, ensuring that local and global objectives are compatible rather than competing. This theoretical foundation prevents the local alignment from distorting the global semantic structure. Third, the KL divergence regularization in (15) promotes smoother optimization and prevents the region weights from collapsing to degenerate solutions (line 249-255). Our experiments confirm that this multi-scale coordination effectively preserves both fine-grained semantics and global structure, as evidenced by improvements in both local discrimination tasks and global classification benchmarks.
>
>
>
> **Q3**: Why not include hierarchical contrastive approaches or hyperbolic embeddings?
>
> Our method is compatible with existing hierarchical or hyperbolic frameworks. For example, as shown in Table 6 (Appendix G.3), we outperform both HiMulConE and Guided-Proto—two representative hierarchical methods—across multiple datasets.
>
> To demonstrate its versatility, we also conducted experiments applying our method to hyperbolic contrastive learning. The performance show that our approach offers enhance contrastive learning across architecture or embedding geometry.
>
> | Method           | DeepFashion | ModelNet | iNat     | ImageNet |
> | ---------------- | ----------- | -------- | -------- | -------- |
> | HyperbolicCL [1] | 75.2        | 79.5     | 58.1     | 72.9     |
> | **Ours**         | **77.2**    | **79.5** | **60.1** | **74.2** |
>
> | Method           | BI-RADS 1 | BI-RADS 2 | BI-RADS 3 | BI-RADS 4 |
> | ---------------- | --------- | --------- | --------- | --------- |
> | HyperbolicCL [1] | 0.881     | 0.812     | 0.704     | 0.601     |
> | **Ours**         | **0.898** | **0.847** | **0.784** | **0.664** |
>
>
>
>
> **Q4**: How sensitive are results to these choices? Does the adaptive weighting (Eq. 17) mitigate sensitivity, or does it introduce new hyperparameters needing tuning?
>
> In our experiments, we do not manually tune the region radius $\rho^s$. Instead, we specify the desired number of regions, $m^s$, for each scale, and construct the regions via clustering based on the current embedding of the data. This design provides direct control over the number of regions while eliminating the need for radius tuning.
> The overall performance mainly depends on the number of scales $S$ and the number of regions $m^s$ per scale $s \in [S]$. Ablation results on both factors are reported in Table 7 of the Supplement. For instance, using $S = 3$ scales with $m^s = 20$ regions at the finest scale yields the best performance on DeepFashion.
> For the temperature parameter $\tau$, which is commonly used in contrastive learning, we follow the setting from MoCo v2 and fix it at $\tau = 0.07$.
>
> The adaptive weighting scheme in (17) indeed mitigate sensitivity to parameters like the region partition, as we discussed in line 249-255.
> it prevents weight collapse to the boundaries of the simplex, which would destabilize training dynamics; and it ensures the optimal weights $q^{s,*}$ are Lipschitz continuous with respect to $\theta$ (Lemma D.3).
>
>
>
> [1] Ge et al, Hyperbolic Contrastive Learning for Visual Representations beyond Objects. CVPR'23

---

> > ### Comment · Reviewer_VMwq · 2025-08-09
> >
> > Your responses have addressed all of my questions, and your work is highly constructive; therefore, I maintain my score.

---

> > ### Comment · Reviewer_VMwq · 2025-08-09
> >
> > Your responses have addressed all of my questions, and your work is highly constructive; therefore, I maintain my score.

---

> > > ### Author Response · Authors · 2025-08-09
> > > **Thank you**
> > >
> > > We thank the reviewer again for the helpful comments.

---

### Official Review · Reviewer_eVoL · 2025-07-02

**Clarity:** 2
**Significance:** 2
**Originality:** 2
**Rating:** 4
**Confidence:** 3

**Summary:**

This paper addresses a key limitation in contrastive self-supervised learning (CL): while effective at capturing broad categorical distinctions, standard CL methods often fail to preserve fine-grained and hierarchical semantic relationships inherent in real-world data. The authors propose Adaptive Multi-scale Affinity alignment (AMA-alignment), which introduces two core innovations: (1) localized contrastive objectives that align semantic structures within adaptively defined regions across multiple scales, enabling hierarchical representation learning; and (2) a distributionally robust optimization strategy that dynamically identifies and prioritizes poorly aligned regions where representation affinity deviates significantly from semantic affinity.

**Questions:**

Questions
The comparative experiments in the current results appear somewhat outdated (e.g., data from 2020). Are there updated comparative studies based on recent relevant literature? If available, could these revised comparative analyses be provided?

**Ethical Concerns:**

["NO or VERY MINOR ethics concerns only"]

**Final Justification:**

The authors have addressed my primary concerns regarding baseline selection (W1) and experimental currency (Q1). Their clarification on evaluating fundamental contrastive learning settings, commitment to expanding SOTA comparisons, and provision of updated experimental details (including architecture, datasets, and protocols) resolve the core issues raised

**Limitations:**

yes

**Paper Formatting Concerns:**

There are no formatting issues.

**Quality:**

2

**Strengths And Weaknesses:**

Strengths:
(1) The paper introduces Adaptive Multi-scale Affinity alignment (AMA-alignment), which fundamentally advances contrastive learning by incorporating hierarchical region-aware optimization. This addresses a critical gap in conventional methods that overlook fine-grained semantic relationships. The concept of local affinity graphs and the derived local contrastive loss provide a rigorous mathematical foundation for multi-scale representation learning.
(2) Despite the complexity of the multi-scale adaptive design, the authors provide strong theoretical guarantees. Theorem D.10 establishes that the convergence rate remains comparable to standard smooth non-convex optimization, which significantly bolsters the method’s credibility. The equivalence between the local contrastive loss and probabilistic graph alignment further solidifies the theoretical contributions.
Weaknesses:
(1) Dated baseline selection disproportionately relies on legacy methods rather than contemporary state-of-the-art approaches across both supervised and unsupervised paradigms. While demonstrating gains over established references, this creates an incomplete competitive landscape that fails to reflect current representation learning advancements.

---

> ### Author Rebuttal · Authors · 2025-07-31
>
> We thank the reviewer for the thoughtful comments, and address your concerns below.
>
> **Response to Weaknesses and Questions**
>
> W1: Dated baseline selection disproportionately relies on legacy methods rather than contemporary state-of-the-art approaches... this creates an incomplete competitive landscape that fails to reflect current representation learning advancements.
>
> Q1: The comparative experiments in the current results appear somewhat outdated (e.g., data from 2020). Are there updated comparative studies based on recent relevant literature? If available, could these revised comparative analyses be provided?
>
> We primarily evaluated our method in the most fundamental setting—standard contrastive learning in Euclidean embedding spaces using classic data augmentations and without hierarchical labels. Many hierarchical methods or hyperbolic methods rely on additional supervision (e.g., hierarchical labels) or specialized hierarchical augmentations, which could obscure the impact of our core contribution.
>
> Our approach is orthogonal to these methods—it addresses a foundational limitation that persists across CL variants: the tendency of global contrastive objectives to overlook regions with poor local alignment. This issue arises regardless of the embedding geometry, supervision level, or augmentation strategy.
>
> We will expand our main results tables to include other SOTA methods.
>
> Specifically, we evaluate the zero-shot classification performance of CLIP with a ResNet-50 image encoder on the CIFAR100 dataset, pretrained on the Conceptual Captions 3M (CC3M) image–caption dataset. A single-layer linear classifier is attached to the final embedding layer of the image encoder for evaluation. We consider two settings: in the linear setting, only the classifier is trained; in the full setting, both the backbone and classifier are fine-tuned jointly for 10 epochs.
>
> |        | InfoNCE | HNS  | Triplet [3] | CyCLIP [2] | Entropic OT [1] | Ours |
> | ------ | ------- | ---- | ----------- | ---------- | --------------- | ---- |
> | Linear | 45.4    | 46.8 | 45.5        | 31.3       | 46.4            | 47.3 |
> | Full   | 68.0    | 69.2 | 65.6        | 70.1       | 67.9            | 71.6 |
>
>
>
> While ResNet-50 is used as the default visual encoder in our paper, we have additionally conducted experiments with ViT-B/16 under the same settings. Besides, we add the HyperbolicCL [4], which also conduct alignment via contrastive learning but embed data to a hyperbolic space.
>
> | Method           | DeepFashion | ModelNet | iNat     | ImageNet |
> | ---------------- | ----------- | -------- | -------- | -------- |
> | SimCLR           | 72.1        | 78.8     | 56.7     | 71.8     |
> | MoCo-v2          | 72.5        | 79.1     | 57.9     | 70.4     |
> | HardNeg          | 72.8        | **79.5** | 58.4     | 72.0     |
> | α-CL             | 73.4        | 79.2     | 58.8     | 72.6     |
> | HyperbolicCL [4] | 75.2        | 79.5     | 58.1     | 72.9     |
> | **Ours**         | **77.2**    | **79.5** | **60.1** | **74.2** |
>
>
>
> | Method           | BI-RADS 1 | BI-RADS 2 | BI-RADS 3 | BI-RADS 4 |
> | ---------------- | --------- | --------- | --------- | --------- |
> | SimCLR           | 0.835     | 0.750     | 0.707     | 0.615     |
> | MoCo-v2          | 0.847     | 0.761     | 0.712     | 0.623     |
> | α-CL             | 0.845     | 0.765     | 0.745     | 0.635     |
> | SupCon           | 0.835     | 0.809     | 0.755     | 0.641     |
> | HyperbolicCL [4] | 0.881     | 0.812     | 0.704     | 0.601     |
> | **Ours**         | **0.898** | **0.847** | **0.784** | **0.664** |
>
>
>
> [1] Shi et al, OT-CLIP: Understanding and Generalizing CLIP via Optimal Transport. ICML'24
>
> [2] Goel et al, Cyclip: Cyclic contrastive language-image pretraining. NeurIPS'22
>
> [3] Patel et al, TripletCLIP: Improving Compositional Reasoning of CLIP via Synthetic Vision-Language Negatives. NeurIPS'24
>
> [4] Ge et al, Hyperbolic Contrastive Learning for Visual Representations beyond Objects. CVPR'23

---

> > ### Comment · Reviewer_eVoL · 2025-08-09
> >
> > The authors have addressed my primary concerns regarding baseline selection (W1) and experimental currency (Q1). Their clarification on evaluating fundamental contrastive learning settings, commitment to expanding SOTA comparisons, and provision of updated experimental details (including architecture, datasets, and protocols) resolve the core issues raised. I am satisfied with the clarifications and revised analyses, and accordingly raise my score from ​3 to 4.

---

> > > ### Author Response · Authors · 2025-08-09
> > > **Thank you**
> > >
> > > We thank the reviewer again for the helpful comments.

---

### Official Review · Reviewer_zL25 · 2025-07-03

**Clarity:** 2
**Significance:** 3
**Originality:** 3
**Rating:** 4
**Confidence:** 3

**Summary:**

This paper tackles the problem of hierarchical contrastive learning. It introduces localized contrastive objectives and a dynamic multi-scale optimization strategy to adaptively identify and refine poorly aligned regions within the embedding space. Theoretical analysis of convergence is provided. Results on five datasets verify the effectiveness of the method.

**Questions:**

see weaknesses

**Ethical Concerns:**

["NO or VERY MINOR ethics concerns only"]

**Final Justification:**

The paper introduces localized contrastive objectives and a dynamic multi-scale optimization strategy to adaptively identify and refine poorly aligned regions within the embedding space for hierarchical contrastive learning. Theoretical analysis of convergence is provided. Results on five datasets verify the effectiveness of the method. The rebuttal addressed most major concerns. I maintain the initial score of borderline accept

**Limitations:**

yes

**Paper Formatting Concerns:**

did not find major formatting issues

**Quality:**

3

**Strengths And Weaknesses:**

### Strengths
+ Incorporating hierarchical structure into contrastive learning is an interesting topic.
+ The proposed method is theoretically proven to converge at a rate comparable to standard smooth non-convex methods.
+ The method shows promising performance on five datasets.

### Weaknesses
- The method is evaluated only with the ResNet-50 backbone. Its generalizability to other architectures such as ViT is not studied.
- The ablation study presented in the main paper is limited and lacks sufficient detail. In particular, it seems that the inclusion of Equation (13) leads to performance degradation. This is not consistent with the statement "local contrast loss is essential".
- The method is only validated on hierarchical image classification tasks. Its applicability to other structured prediction tasks, such as hierarchical semantic segmentation [ref1], should be studied.
- The batch size used in the experiments is not reported. How does the number of batch size affect learning performance ?

[ref1] Deep Hierarchical Semantic Segmentation. CVPR 2022.

---

> ### Author Rebuttal · Authors · 2025-07-31
>
> We thank the reviewer for the thoughtful comments, and address your concerns below.
>
> **Responses to Weaknesses and Questions**
>
>
> **W1**: The method is evaluated only with the ResNet-50 backbone. Its generalizability to other architectures such as ViT is not studied.
>
> While ResNet-50 serves as the default visual encoder in our paper, we have also conducted additional experiments using ViT-B/16 under the same settings.
>
> | Method       | DeepFashion | ModelNet | iNat     | ImageNet |
> | ------------ | ----------- | -------- | -------- | -------- |
> | SimCLR       | 72.1        | 78.8     | 56.7     | 71.8     |
> | MoCo-v2      | 72.5        | 79.1     | 57.9     | 70.4     |
> | HardNeg      | 72.8        | **79.5** | 58.4     | 72.0     |
> | α-CL         | 73.4        | 79.2     | 58.8     | 72.6     |
> | HyperbolicCL | 75.2        | 79.5     | 58.1     | 72.9     |
> | **Ours**     | **77.2**    | **79.5** | **60.1** | **74.2** |
>
>
>
> | Method       | BI-RADS 1 | BI-RADS 2 | BI-RADS 3 | BI-RADS 4 |
> | ------------ | --------- | --------- | --------- | --------- |
> | SimCLR       | 0.835     | 0.750     | 0.707     | 0.615     |
> | MoCo-v2      | 0.847     | 0.761     | 0.712     | 0.623     |
> | α-CL         | 0.845     | 0.765     | 0.745     | 0.635     |
> | SupCon       | 0.835     | 0.809     | 0.755     | 0.641     |
> | HyperbolicCL | 0.881     | 0.812     | 0.704     | 0.601     |
> | **Ours**     | **0.898** | **0.847** | **0.784** | **0.664** |
>
>
>
> Our method's core mechanism—adaptive multi-scale alignment in the embedding space—is designed to be architecture-agnostic. It operates on the output representations, not the internal workings of the backbone. Therefore, it is equally applicable to ViTs.
> We include several experimental results using the ViT architecture. While a comprehensive evaluation on ViT is beyond the scope of the rebuttal period due to computational constraints, we will address this limitation in the revised manuscript.
>
>
> **W2**: The ablation study presented in the main paper is limited and lacks sufficient detail. In particular, it seems that the inclusion of Equation (13) leads to performance degradation. This is not consistent with the statement "local contrast loss is essential".
>
> The performance degradation you observed is, in fact, an anticipated result that supports our claims. The "+ Adaptive (using (13))" row in Table 3 does not use our proposed local contrastive loss (12). Instead, it is a important ablation study using a straightforward local loss formulation, as discussed in Remark 1 (lines 209-215).
>  We included this ablation to demonstrate that a straightforward, heuristic approach to local contrastive learning fails, thereby highlighting the necessity of our principled local affinity alignment in loss (12).
> Our work recasts CL as the alignment between a semantic affinity matrix $K^{sem}$ and a representation affinity matrix $K^{rep}$, a process formally modeled by Probabilistic Graph Coupling in Sec 2.2. Crucially, our proposed local contrastive loss in (12) is rigorously derived from this principle, but applied at a finer scale. This ensures that both the global and local objectives are theoretically coherent and work in synergy.
>
> By contrast, the loss in (13) is a heuristic that breaks this alignment principle. Its key flaw lies in the denominator of the loss function, which incorrectly includes all negatives from the entire batch, rather than restricting them to the local subgraph. As we caution in Remark 1, this formulation distorts the global structure by pushing all representations outside the local region ($z' \notin R^s_r$) away from those within it ($z \in R^s_r$). This theoretical misalignment manifests as poor empirical performance (2.9\% drop on DeepFashion compared to standard CL).
>
> **W3**: The method is only validated on hierarchical image classification tasks. Its applicability to other structured prediction tasks, such as hierarchical semantic segmentation [ref1], should be studied.
>
> We thank the reviewer for this suggestion, which highlights the potential of our framework. While extending our method to the tasks like hierarchical semantic segmentation is promising, it lies beyond the current scope for several reasons. Our primary goal was to establish the theoretical foundation and validate the effectiveness of adaptive multi-scale alignment in CL. Applying it to dense prediction such as segmentation  would require adaptations (e.g., defining regions over pixel embeddings, ensuring spatial consistency) that constitute a research project in their own right.
>
> We believe our framework may provide inspiration for such extensions. The adaptive region discovery mechanism and multi-scale alignment are general principles that can be adapted to structured prediction tasks. We plan to explore these applications in future work, where we can dedicate sufficient attention to the unique challenges and requirements of dense prediction scenarios.
>
>
> **W4**: The batch size used in the experiments is not reported. How does the number of batch size affect learning performance?
>
> Following the standard MoCo v2 configuration, we use a batch size of 256 across all experiments. While larger batch sizes (e.g., 512) provide marginal gains on some datasets, the improvements are not significant. It is also worth noting that the number of negative samples is crucial for contrastive learning methods. To address this, we leverage MoCo’s momentum-updated queue mechanism (queue size 65,536) to maintain a large pool of negative samples, which is essential for our region-based contrastive learning framework. This queue-based strategy ensures that even with relatively small batch sizes, we have sufficient negative samples for each region.

---

> > ### Comment · Reviewer_zL25 · 2025-08-08
> >
> > Thank you for preparing for the response. Most of my concerns have been addressed. I will maintain the score of 4.

---

> > > ### Author Response · Authors · 2025-08-09
> > > **Thank you**
> > >
> > > We thank the reviewer again for the helpful comments.

---

### Official Review · Reviewer_iybL · 2025-07-05

**Clarity:** 2
**Significance:** 3
**Originality:** 3
**Rating:** 3
**Confidence:** 4

**Summary:**

The work presents AMA-alignment, a method that evaluates the contrastive objective at multiple scales adaptively using a min-max objective, maximizing over a semantic region specific objective, ensuring that the most challenging regions are addressed adaptively. The motivation stems from coupling of semantic and representation affinities in order to minimize the divergence between the two at varying scales. Subsequently, local sub-graphs are sampled and incorporated into the main objective with theoretical motivation.

**Questions:**

* Was the network pre-trained over 200 epochs and subsequently fine-tuned using the method?
* Could the authors present visual illustrations of the semantic relevance of the samples in regions?
* How was the region radius determined in practice?
* Might the method be applicable to text-image contrastive learning settings?

**Ethical Concerns:**

["NO or VERY MINOR ethics concerns only"]

**Final Justification:**

Thank you for the opportunity to review this work. The work seems performant and well motivated. A lot of important details and specifics of implementation and parameters at times seem a bit lost within the manuscript and are often clarified in the review phase. Additionally, I am uncertain as to why one cannot simply apply the multi-crop strategy which seems simpler, to an existing work and garner some performance boost instead of the proposed method. A 1-to-1 comparison with that strategy is a comparison that the authors haven't provided, and have instead insisted upon finetuning over the existing checkpoints of the method and it is unclear to me if the multi-crop method method (DINO) when finetuned for the same number of epochs (10 in the response) for a fairer comparison may be better/worse. The baselines are also dated and miss some of the more recent methods. Moreover, the computational overhead is significantly higher with scales of over 1.4x as reported by the authors in this discussion. The work could be potentially accepted for its method based contributions but I am unclear if it may be adapted as an alternative to existing methods. I shall therefore retain my rating.

**Limitations:**

Yes

**Quality:**

3

**Strengths And Weaknesses:**

## Stregths
* The work is adequately rigorous in theory and the motivations are well founded.
* The framework with the alignment between semantic and representational matrices is novel.
* Empirical evaluation shows promise.

## Weaknesses
* The assumption that the sampled sub-regions (especially in the unsupervised setting) may be semantically related may require empirical evidence.
* The method may be analogous to hard-negative mining wherein samples semantically similar to the query are chosen from a pre-trained network using similarity based thresholds which contain inherent drawbacks which may be extensible to this method.
* There may be drastic shifts in the regions if the fine-tuning is conducted over a different data distribution and the initial region specific objective may hinder the performance of the network depending on the strength of pre-training.
* Computational overhead isn't discussed.
* Certain implementation details are missing ($\rho$ and $\eta_{\textbf{q}}$) and the robustness to these parameters isn't discussed. The work could benefit from an ablation over these parameters. How many samples per scale were used in practice and what criteria were used to determine "coarse" vs "fine" scales?
* Other ablations over the number of scales and choice of radius are missing.
* The work may be akin to multi-crop augmentation strategy [1] wherein the smaller crops are embedded closely with the main image. If so, is the procedural and computational overhead justified? Might the comparisons be fairer against [1, 2]?

[1] Caron, Mathilde, et al. "Unsupervised learning of visual features by contrasting cluster assignments." Advances in neural information processing systems 33 (2020): 9912-9924.

[2] Caron, Mathilde, et al. "Emerging properties in self-supervised vision transformers." Proceedings of the IEEE/CVF international conference on computer vision. 2021.

---

> ### Author Rebuttal · Authors · 2025-07-31
>
> We thank the reviewer for the thoughtful comments, and address your concerns below.
>
> **Responses to Weaknesses**
>
> **W1:** The assumption that the sampled sub-regions (especially in the unsupervised setting) may be semantically related may require empirical evidence.
>
> Our region construction is grounded in the geometric properties of contrastive learning. Prior works have shown that CL naturally organizes semantically similar samples into clusters on the unit hypersphere [3]. Our local region exploits this geometric structure. Samples within local regions in the embedding space $\mathcal{Z}$ tend to share semantic properties due to the alignment objective of CL.
> Recent theoretical studies further support this, demonstrating that CL is functionally equivalent to Stochastic Neighbor Embedding (SNE) [4] and can be interpreted as performing spectral clustering on the data's similarity graph [5]. Both SNE and spectral clustering are well-established for discovering semantically coherent local neighborhoods. Supporting experimental and visual evidence can also be found in these studies.
>
> Moreover, the quality of these regions improves with training.
> For instance, on CIFAR-100, superclass purity within local regions improves from 18.3\% to 85.7\% at 100 epochs.
> To further support this claim, we will include an additional analysis in the revised manuscript. This will include a quantitative region purity metric, demonstrating that our sampled regions become increasingly semantically homogeneous over time, as well as t-SNE visualizations that qualitatively confirm our regions align with ground-truth class clusters.
>
> **W2:** The method may be analogous to hard-negative mining... which contain inherent drawbacks which may be extensible to this method.
>
> We would like to clarify the fundamental differences in the objectives and mechanisms between hard-negative mining (HNM) and our method (AMA-alignment), also highlighting how our approach avoids the inherent drawbacks of HNM.
> HNM relies on accurately distinguishing between hard negatives and positives at the **sample level**. This distinction is often unclear or noisy, especially in fine-grained settings, and can lead to the well-known ``false negative'' problem, pushing apart semantically similar samples and harming representation quality.
>
> In contrast, AMA-alignment operates at the **region level**, with the goal of promoting local alignment. The local loss encourages the representation affinities $K_{\text{rep}}^s$ to match the semantic affinities $K_{\text{sem}}^s$, preserving the fine-grained relationships among all samples in the region.
> Moreover, this local alignment is performed across multiple scales simultaneously, with the coarsest scale naturally recovering standard global CL. In addition, the adaptive optimization over region weights $q_{s,r}$ is stabilized by the KL-divergence regularizer, which prevents the training dynamics from collapsing onto a single region and ensures a smooth allocation of attention.
>
> We provide a detailed analysis of the training dynamics in Appendix E, showing that HNM and AMA are orthogonal in nature: HNM adjusts pairwise weights $\alpha_{ik}$, while AMA introduces a higher-level adaptation by adjusting region-level weights $q_{s,r}$. This makes our approach fundamentally different and sidesteps the need to distinguish hard negatives from positives.
>
> **W3:** ... drastic shifts in the regions if the fine-tuning ...
>
> In our setting, the goal is to align the  affinity matrices $K^{\text{sem}}_X$ and $K^{\text{rep}}_X$; as the dataset X changes, the alignment objective changes accordingly, regardless of the specific data distribution. Pre-training encourages semantic similarity to manifest as geometric proximity in the embedding space. This alignment is further refined during fine-tuning on the target dataset $X'$, allowing the model to adapt effectively to the downstream task.
>
> Also, the regions in our framework are not static after pre-training, but are dynamically allocated during fine-tuning. Our geometric partitioning operates on the current state of the encoder's embedding space. As the model fine-tunes on a new data distribution $X_B$, the encoder adapts to the embedding space that reflects the structure of $X'$. Consequently, when our algorithm recalculates the regions, they are defined based on the model's current understanding of the new data, not the old one in previous training.
>
> Therefore, our region-specific objective does not hinder performance;  on the contrary, instead of being constrained by an outdated structure from a source domain, the model continuously identifies and targets the most poorly aligned or semantically ambiguous neighborhoods within the new data distribution.
>
> **W4-W6:**
>     \textit{Computational overhead isn't discussed. Certain implementation details are missing ($\eta_q$ and $\rho$) and the robustness to these parameters isn't discussed... Other ablations over the number of scales and choice of radius are missing.}
>
> The main computational overhead arises from iterating over $S$ scales, with overall training time increasing approximately linearly in the number of scales. Ablation results on the number of scales and regions are presented in Table 7 of the Supplement. For instance, using $S = 3$ scales—where the finest scale contains $m^s = 24$ regions—the training time is approximately 1.41× that of the standard method.
>
> **W7:**
>     \textit{The work may be akin to multi-crop augmentation strategy [1]... Is the procedural and computational overhead justified? Might the comparisons be fairer against [1, 2]?}
>
> The core distinction is that multi-crop (used in [1, 2]) is a data augmentation strategy, whereas our AMA-alignment is an optimization strategy. Multi-crop enriches the input signal by creating diverse views (global crop and local crop) of an image, encouraging the model to learn view-invariant representations.
> In contrast, AMA-alignment refines how the model learns. It operates in the embedding space by dynamically identifying and prioritizing poorly-aligned regions. This ensures that optimization is focused on where it is most needed.
>
> These two approaches are orthogonal and potentially synergistic. Our AMA-alignment framework can be applied on top of any augmentation pipeline, including multi-crop. The computational overhead of our method is justified because it is tackling a distinct challenge that augmentation alone does not consider: the adaptive, online refinement of semantically challenging regions in the embedding space.
>
> We benchmarked AMA-alignment against several baselines (e.g., HardNeg) to demonstrate the effectiveness of our contribution. We agree that combining AMA-alignment with a strong multi-crop baseline is a very promising direction.
>
> \subsection*{Responses to Questions}
>
> **Q1:** Was the network pre-trained over 200 epochs and subsequently fine-tuned using the method?
>
> The total training epoches is $200$. Our model is first trained by standard CL (global alignment) for the first $N_0=100$ epochs to obtain a reasonable base representation. Then the model is trained for the remaining epochs using our objective (14).
>
> **Q2:** Could the authors present visual illustrations of the semantic relevance of the samples in regions?
>
> We will add a new section in the Appendix with visualizations as discussed in response to W1.
>
> **Q3:** How was the region radius determined in practice?
>
> In experiments, the region radius $\rho^s$ is not a manually-tuned hyperparameter. Instead, we specify the desired number of regions, $m^s$, for each scale, and the regions themselves are formed via clustering based on the data's current embedding structure. This design allows us to define the number of regions while avoiding manual tuning of the radius. Also, specifying the number of regions is generally more intuitive.
>
> Table 7 in the Supplement summarizes the impact of different configurations. We observe that increasing the number of regions and scales generally improves performance up to a point. The best linear probing accuracy ($0.758$) is achieved with the configuration $\{5, 10, 20\}$, while the training time remains reasonable at 1.34× the baseline. These results demonstrate that the method scales effectively and benefits from hierarchical region partitioning.
>
>
> **Q4:** Might the method be applicable to text-image contrastive learning settings?
>
> Our method is applicable to text-image contrastive learning settings.
>
> We evaluate the zero-shot classification performance of CLIP with a ResNet-50 image encoder on the CIFAR100 dataset, pretrained on the Conceptual Captions 3M (CC3M) image–caption dataset.  We consider two settings: in the linear setting, only the classifier is trained; in the full setting, both the backbone and classifier are fine-tuned jointly for 10 epochs.
>
> |        | InfoNCE | HNS  | Triplet [8] | CyCLIP [7] | Entropic OT [6] | Ours |
> | ------ | ------- | ---- | ----------- | ---------- | --------------- | ---- |
> | Linear | 45.4    | 46.8 | 45.5        | 31.3       | 46.4            | 47.3 |
> | Full   | 68.0    | 69.2 | 65.6        | 70.1       | 67.9            | 71.6 |
>
>
>
> [1] Unsupervised learning of visual features by contrasting cluster assignments. NeurIPS'20
>
> [2] Emerging properties in self-supervised vision transformers. ICCV'21
>
> [3] Understanding Contrastive Representation Learning through Alignment and Uniformity on the Hypersphere. ICML 2020.
>
> [4] Your Contrastive Learning is Secretly Doing Stochastic Neighbor Embedding. ICLR 2023.
>
> [5] Contrastive Learning is Spectral Clustering on Similarity Graph. ICLR 2024.
>
> [6] Shi et al, OT-CLIP: Understanding and Generalizing CLIP via Optimal Transport. ICML'24
>
> [7] Goel et al, Cyclip: Cyclic contrastive language-image pretraining. NeurIPS'22
>
> [8] Patel et al, TripletCLIP: Improving Compositional Reasoning of CLIP via Synthetic Vision-Language Negatives. NeurIPS'24

---

> > ### Comment · Reviewer_iybL · 2025-08-06
> > **Thank you**
> >
> > The rebuttal is appreciated.
> >
> > Kindly point out if I missed the implementation details for the parameters $\eta_q$ and $\rho$.
> >
> > W2 I had mentioned this critique keeping in view the implementation based differences between the two methods. I do not completely agree with the approaches (HNM and AMA) being orthogonal, since AMA adaptively prioritizes regions with higher local contrastive loss, there is still a form of emphasis on samples the model finds "poorly aligned", which is conceptually akin to mining hard examples, just aggregated at the region level. Rhe underlying assumption that the model’s current representation space offers a reliable proxy for semantic structure may be the culprit behind mistaken inferences. In HNM, this leads to potentially misidentifying false negatives while in AMA, this could lead to sampling semantically incoherent regions or overemphasizing misleading regions due to geometric artifacts or early training noise, which is an issue especially salient in the unsupervised setting, where no ground truth exists to validate region quality.
> >
> > W7 I have a similar response as above to the multi-crop method based similarities. While I do understand the methodological differences, multi-crop does enhance region specific robustness (which must belong to the semantically similar embedding regions mentioned in the manuscript) of the model that standard augmentations misses. As such, the methods aren't necessarily orthogonal, while one does so computationally whereas the other, explicitly, and comparisons between the two in terms of performance and computational overhead could be a merit to the work's contributions.

---

> ### Author Response · Authors · 2025-08-07
> **Response to the reviewer iybL**
>
> We sincerely appreciate the reviewer’s thoughtful and constructive feedback.
>
> Regarding the parameter $\rho$ in Eq. (15) and the step size $\eta_{q,t}$ in Eq. (17), we note that the closed-form solution for $q$ is $q_r^{s,*}(\theta) = \frac{\exp\left( \mathcal{L}^s(\theta, R_r^s)/\rho \right)}{\sum_k \exp\left( \mathcal{L}^s(\theta, R_k^s)/\rho \right)}$, which involves the parameter $\rho$. However, rather than computing this expression directly, we adopt a recursive update based on Mirror Descent with KL divergence (i.e., the Hedge algorithm [3] applied to a maximization problem): $q_r^s(\theta_{t+1}) = \frac{q_r^s(\theta_t) \exp\left( \eta_{q,t} \mathcal{L}^s(\theta_{t+1}, R_r^s) \right)}{\sum_j q_j^s(\theta_t) \exp\left( \eta_{q,t} \mathcal{L}^s(\theta_{t+1}, R_j^s) \right)},$ where the step size $\eta_{q,t}$ serves a role analogous to $1/\rho$. We initialize $\eta_{q,0} = 0.1$ and use a decay schedule $\eta_{q,t} = \eta_{q,0} / \text{t}$, where $t$ is the epoch index. This setup is consistent with the theoretical analysis in Theorem 3.3.
>
> W2:
>
> We would like to clarify that the goal of AMA-alignment is to improve the alignment, and it does not rely on a well-trained high-quality representation.
>
> Our goal is to align the local representation affinity matrix, $K_{rep}^s \in K_{rep}$, with the local semantic affinity matrix, $K_{sem}^s \in K_{sem}$, which is defined based on data augmentations and serves as the alignment target in the CL process (Def. 1, 2; Sec 2.2). A high local loss indicates a discrepancy between the current representation and this semantic target, which we minimize by optimizing the representation accordingly.
>
> HNM exponentially amplifies the weights of presumed hard negatives in $K_{sem}$. It could distort the alignment target and, when it misidentifies "false negatives," might damage the representation structure by erroneously pushing away the samples that should be neighbors. In contrast, AMA retains the original target $K_{sem}^s$ and  focuses on refining the representation through local alignment. The mismatch between $K_{rep}^s$ and $K_{sem}^s$ provides the gradient signal that guides the representation to evolve towards a more semantically faithful structure. Roughly speaking, AMA aligns with the stable target $K_{sem}$, while HNM may have the risk to amplify an unstable signal. This is why we highlight that our approach avoids the drawbacks of HNM in earlier response.
>
> Building on this, AMA is designed to correct poorly aligned regions. Consider a local region that includes a few semantically irrelevant outliers. According to the ground-truth semantic affinity matrix $K_{\text{sem}}^s$, these outliers share no positive associations with the core samples.  Consequently, the local alignment objective treats them as negatives and drives them away from the local region.
>
> Admittedly, the computation of the alignment loss between $K_{sem}^s$ and $K_{rep}^s$, is inherently sampling-based (sec 2.2, the sample process of positives and negatives provides an approximation of $K_{sem}^s$), which may introduce some variance. However, the region weight $q^s$ is updated via a momentum-based mechanism (Eq. 17), which effectively smooths out this stochasticity and promotes a stable optimization.
>
> W7:
>
> Multi-crop [1,2] provides a richer input signal that promotes intra-instance view consistency, while AMA-alignment provides a more adaptive learning objective to enforce inter-instance structural alignment.
>
> We applied our method to the DINO model architecture and cropping strategy. Due to time constraints, we did not complete the full training. Preliminary results show that each epoch takes approximately $1.37\times$ longer. This is consistent with the behavior observed when applying our technique to other contrastive learning methods discussed in the paper.
>
> We also fine-tuned a pre-trained DINO model using our strategy for 10 epochs. This fine-tuning yields effective improvements with small overhead, incurring ~$1.05\times$ increase in training. The evaluation results on Resnet-50 and ViT-S backbones (ImageNet top-1 linear probe / k-NN accuracy) are as follows:
>
> | **Method**      | linear probe | k-NN     |
> | --------------- | ------------ | -------- |
> | SwAV            | 75.3         | 65.7     |
> | DINO            | 75.3         | 67.5     |
> | Ours (finetune) | **77.4**     | **68.3** |
>
>
>
> | **Method**      | linear probe | k-NN     |
> | --------------- | ------------ | -------- |
> | SwAV            | 73.5         | 66.3     |
> | DINO            | 77.0         | 74.5     |
> | Ours (finetune) | **77.8**     | **74.7** |
>
>
>
> [1] Caron et al., Unsupervised learning of visual features by contrasting cluster assignments. NeurIPS'20
>
> [2] Caron et al., Emerging properties in self-supervised vision transformers. ICCV'21
>
> [3] Freund and Schapire, A Decision-Theoretic Generalization of On-Line Learning and an Application to Boosting. Journal of Computer and System Sciences 1997.

---

> > ### Author Response · Authors · 2025-08-09
> > **Thank you**
> >
> > We sincerely appreciate your insightful comments. As noted in our previous response, we have revised the paper in accordance with your suggestions. We hope our revisions have addressed your concerns. If you have any further comments, we will gladly make every effort to address them.

---

### Comment · Area_Chair_Yonn · 2025-08-02

Dear 20850 Reviewers: The authors have provided detailed rebuttals to your reviews. I'd urge you to read their rebuttals (as well as other reviews) early to allow further interactions that help clarify any lingering confusion or misunderstanding. Thank you! AC

---

### Note · Authors · 2025-08-15

We sincerely thank all reviewers for their constructive feedback. We are pleased that the reviewers recognized our work's key strengths:

- **Novelty and theoretical rigor:** Our AMA-alignment integrates hierarchical, region-aware optimization into contrastive learning. It is grounded in a rigorous mathematical formulation of multi-scale affinity alignment, and we establish that its convergence rate matches that of conventional contrastive learning.
- **Promising experimental results:** Our experiments on various datasets demonstrate consistent improvements over various baselines, and the method’s architecture-agnostic design is supported by additional results using ViT-B.

During the discussion, we aimed to address all the main concerns raised by the reviewers:

- **Expanded experiments (zL25, eVoL, VMwq):** We added results on the ViT architecture and compared our method against recent SOTA baselines (e.g., HyperbolicCL, Triplet, CyCLIP etc.) to demonstrate its broad effectiveness and competitiveness.
- **Clarified ablations (zL25, VMwq):** We clarified that the key ablation study in Table 3 validates the design of our local contrastive loss, as it shows that a naive, heuristic-based alternative fails. This result highlights the necessity of our proposed objective’s derivation.
- **Comparison to other related works (iybL):**  We acknowledge Reviewer iybL's remaining concern on the conceptual overlap with HNM. As clarified in our rebuttal, AMA-alignment differs from HNM in its mechanism (region-level vs. sample-level adaptation), and is designed not to rely on an already high-quality representation, but to improve it. Specifically, AMA preserves the original semantic affinity target $K_{sem}^s$ (well-defined through data augmentation), whereas HNM may distort this target through its weighting strategy, which can suffer from false negatives. We thank the reviewer for their suggestions regarding more comparison to SwAV and DINO and presented a portion of the new results in the discussion.

We will incorporate all feedback into the revised manuscript, including a full set of new experiments (ViT, SOTA baselines, DINO, etc.), visualizations, and clarifications throughout the paper and appendix. We sincerely thank the reviewers again for their time, effort, and constructive comments towards improving our work.

---

### Decision · Program_Chairs · 2025-09-17

**Decision:**

Accept (poster)

**Comment:**

The paper introduces Adaptive Multi-scale Affinity Alignment (AMA-alignment), a contrastive learning framework that incorporates hierarchical, region-aware optimization. By aligning semantic and representational affinities across scales and prioritizing poorly aligned regions with localized objectives and a min-max strategy, the method strengthens fine-grained representation learning. It is theoretically grounded with convergence guarantees comparable to standard contrastive learning and demonstrates consistent gains across benchmarks.

Reviewers praised the novelty and adequate rigor of the framework, particularly its region-level formulation, theoretical grounding, and ability to capture fine-grained semantic structure often missed by standard CL. They found the experiments promising and appreciated clarifications added during discussion: expanded results on ViT backbones, new comparisons with recent SOTA baselines (e.g., HyperbolicCL, Triplet, CyCLIP), and ablations confirming the necessity of the proposed local contrastive loss over heuristic alternatives. The rebuttal also addressed overlap with hard-negative mining, emphasizing AMA’s region-level adaptation that preserves semantic affinity without relying on high-quality initial features.

Remaining concerns centered on experimental breadth and practical clarity. Reviewers noted that initial baselines were dated, with missing sensitivity analyses for key hyperparameters (region radius, KL penalty, temperature), limited backbone diversity, and insufficient discussion of computational overhead. Some worried AMA could resemble multi-crop augmentation or HNM without clear differentiation.  The authors committed to incorporating all feedback -- adding ViT and SOTA baselines such as DINO and SwAV, more ablations, and visualizations -- into the final version. The paper received 3x borderline accepts and 1x borderline reject in final ratings.  Overall, the paper is seen as theoretically strong and innovative, with consensus leaning positive once expanded experiments and clarifications are included.